# Characteristics of Unripened Cow Milk Curd Cheese Enriched with Raspberry (*Rubus idaeus*), Blueberry (*Vaccinium myrtillus*) and Elderberry (*Sambucus nigra*) Industry By-Products

**DOI:** 10.3390/foods12152860

**Published:** 2023-07-27

**Authors:** Vytaute Starkute, Justina Lukseviciute, Dovile Klupsaite, Ernestas Mockus, Jolita Klementaviciute, João Miguel Rocha, Fatih Özogul, Modestas Ruzauskas, Pranas Viskelis, Elena Bartkiene

**Affiliations:** 1Department of Food Safety and Quality, Faculty of Veterinary, Lithuanian University of Health Sciences, Tilzes Str. 18, LT-47181 Kaunas, Lithuania; vytaute.starkute@lsmu.lt (V.S.); justina.lukseviciute@lsmu.lt (J.L.); 2Faculty of Animal Sciences, Institute of Animal Rearing Technologies, Lithuanian University of Health Sciences, Tilzes Str. 18, LT-47181 Kaunas, Lithuania; dovile.klupsaite@lsmu.lt (D.K.); ernestas.mockus@lsmu.lt (E.M.); jolita.klementaviciute@lsmu.lt (J.K.); 3Universidade Católica Portuguesa, CBQF—Centro de Biotecnologia e Química Fina—Laboratório Associado, Escola Superior de Biotecnologia, Rua Diogo Botelho 1327, 4169-005 Porto, Portugal; jmfrocha@fc.up.pt; 4Laboratory for Process Engineering, Environment, Biotechnology and Energy (LEPABE), Faculty of Engineering, University of Porto (FEUP), Rua Dr. Roberto Frias, s/n, 4200-465 Porto, Portugal; 5Associate Laboratory in Chemical Engineering (ALiCE), Faculty of Engineering, University of Porto, Rua Dr. Roberto Frias, s/n, 4200-465 Porto, Portugal; 6Department of Seafood Processing Technology, Faculty of Fisheries, Cukurova University, Balcali, Adana 01330, Turkey; fozogul@cu.edu.tr; 7Biotechnology Research and Application Center, Cukurova University, Balcali, Adana 01330, Turkey; 8Department of Anatomy and Physiology, Faculty of Veterinary, Lithuanian University of Health Sciences, Tilzes Str. 18, LT-47181 Kaunas, Lithuania; modestas.ruzauskas@lsmu.lt; 9Faculty of Veterinary, Institute of Microbiology and Virology, Lithuanian University of Health Sciences, Tilzes Str. 18, LT-47181 Kaunas, Lithuania; 10Lithuanian Research Centre for Agriculture and Forestry, Institute of Horticulture, Kauno Str. 30, LT-54333 Babtai, Lithuania; pranas.viskelis@lammc.lt

**Keywords:** unripened cow milk curd cheese, berry industry by-product, antimicrobial properties, antioxidant characteristics, biogenic amine, volatile compounds, nutrition

## Abstract

The aim of this study was to apply raspberry (Ras), blueberry (Blu) and elderberry (Eld) industry by-products (BIB) for unripened cow milk curd cheese (U-CC) enrichment. Firstly, antimicrobial properties of the BIBs were tested, and the effects of the immobilization in agar technology on BIB properties were evaluated. Further, non-immobilized (_NI_) and agar-immobilized (_AI_) BIBs were applied for U-CC enrichment, and their influence on U-CC parameters were analyzed. It was established that the tested BIBs possess desirable antimicrobial (raspberry BIB inhibited 7 out of 10 tested pathogens) and antioxidant activities (the highest total phenolic compounds (TPC) content was displayed by _NI_ elderberry BIB 143.6 mg GAE/100 g). The addition of BIBs to U-CC increased TPC content and DPPH^−^ (2,2-diphenyl-1-picrylhydrazyl)-radical scavenging activity of the U-CC (the highest TPC content was found in C-Ra_NI_ 184.5 mg/100 g, and strong positive correlation between TPC and DPPH^−^ of the U-CC was found, r = 0.658). The predominant fatty acid group in U-CC was saturated fatty acids (SFA); however, the lowest content of SFA was unfolded in C-Eld_AI_ samples (in comparison with C, on average, by 1.6 times lower). The highest biogenic amine content was attained in C-Eld_AI_ (104.1 mg/kg). In total, 43 volatile compounds (VC) were identified in U-CC, and, in all cases, a broader spectrum of VCs was observed in U-CC enriched with BIBs. After 10 days of storage, the highest enterobacteria number was in C-Blu_NI_ (1.88 log_10_ CFU/g). All U-CC showed similar overall acceptability (on average, 8.34 points); however, the highest intensity of the emotion “happy” was expressed by testing C-Eld_NI_. Finally, the BIBs are prospective ingredients for U-CC enrichment in a sustainable manner and improved nutritional traits.

## 1. Introduction

Cheese, including unripened cow milk curd cheese (U-CC), is a very popular product in many countries around the world. In 2021, consumption of cheese in the European Union (EU) was, on average, 20.4 kg per person; in the United States of America and Canada it was, on average, 17.9 and 15.0 kg per person, respectively [1]. There are many types of cheese [2], and their characteristics differ in relation with technology used for their preparation. In Eastern Europe, curd cheese and unripened cow milk curd cheese (U-CC) are very popular daily food products, and the dairy industry is looking for natural additives to enrich them and to chase innovation, higher added-value and improvement of food quality, food safety and nutritional value [3]. The most popular ingredients in U-CC production are caraway seeds; however, it was reported that licorice root (*Glycyrrhiza glabra*) can be an attractive supplement, which could improve U-CC sensory properties, fatty acid (FA) and volatile compound (VC) profiles [4]. Additionally, enrichment of cheese with cranberry extract can significantly improve the antimicrobial properties of the end product [5]. Notwithstanding, U-CC has a problem due to its relatively short shelf-life, but this can be controlled by including natural antimicrobial compounds to the main cheese formula [6,7]. Hence, there are many possibilities to enrich such types of cheese with functional compounds, and one of these possibilities is to include berry industry by-products (BIBs).

Industrial processes (juice, wine, etc. production) greatly contribute to BIB production, which, until now, has not been used efficiently enough [8]. In this research study, we hypothesized that raspberry (Ras), blueberry (Blu) and elderberry (Eld) production by-products can be employed in high-added value (better antioxidant properties and higher sensory acceptability, due to a higher variety of volatile compounds, etc.) U-CC preparation.

The raspberry (*Rubus idaeus*) processing industry generates huge amounts of by-products that are rich in bioactive compounds [9] and can be used as valuable ingredients to improve other food products (especially of animal origin) with functional characteristics, resulting from enriching them with natural antioxidants, anthocyanins, flavonoids, phytochemicals, carotenoids, polyphenols, vitamins and minerals [9].

Blueberries (*Vaccinium myrtillus*) are popular berries with an attractive flavor and numerous health benefits [10] which have been attributed to an important number of bioactive compounds in these berries [10]. Additionally, blueberries are an abundant source of sugars (glucose and fructose), vitamins, folic acid, minerals, organic acids, flavanols and anthocyanins [11,12,13,14,15].

Elderberries (*Sambucus nigra*) are rich in many bioactive compounds, such as polyphenolic and terpenoid compounds and anthocyanins [16]. For this reason, elderberries are included in many food formulations [17]. Elderberries’ beneficial effect on human health is widely described [16,18,19].

Therefore, the incorporation of BIB into the main U-CC formula could be very attractive. However, the latter ingredients, due to their complex composition, could possess several effects on U-CC quality and safety, including non-desirable ones. Taking into consideration that BIBs are rich in colored compounds and some of them are not stable, they can change the color of the product during the U-CC preparation process. Color is one of the main sensory characteristics, and, for this reason, ensuring an attractive color for consumers is a very important issue [20]. Aiming to reduce contact between incorporated BIBs with other U-CC ingredients, agar-immobilization technology for BIB was tested in this study. Agar is known as a food ingredient possessing mucoadhesive properties, high firmness and a desirable end-product texture [21,22,23].

Finally, the aim of this study was to apply BIBs for U-CC enrichment. To implement such a goal, a two-stage experiment was performed. During the first stage, antimicrobial properties of the BIBs were tested, and the effects of the immobilization technology on BIB antioxidant properties and color coordinates were evaluated. During the second stage, non-immobilized (_NI_) and agar-immobilized (_AI_) BIBs were applied to U-CC enrichment, and their influence on U-CC acidity parameters, color characteristics, moisture content, VC and FA profiles, biogenic amine (BA) concentration, sensory properties, induced emotions for consumers and microbiological characteristics during the storage were analyzed.

## 2. Materials and Methods

### 2.1. Materials Used for Berry Industry By-Product (BIB) Immobilization, Unripened Cow Milk Curd Cheese (U-CC) Preparation and General Experiment of the Current Study

Raspberry (*Rubus idaeus*, variety ‘Poliana’), blueberry (*Vaccinium myrtillus*) and elderberry (*Sambucus nigra*) BIBs were obtained after juice production and, consisting of the peels, seeds and fibers, were given by the Institute of Horticulture, Lithuanian Research Centre for Agriculture and Forestry (Babtai, Kaunas distr., Lithuania) in 2023. Before use, the BIBs were vacuum-dried in a vacuum dryer XF020 (France-Etuves, Chelles, France) at 45 ± 2 °C and a pressure of 6 × 10^−3^ mPa and milled by using an ultra-centrifugal mill “ZM 200” (Retsch, Haan, Germany) until a particle size <1 mm was obtained.

For by-product immobilization, agar powder (produced from *Gelidium sesquipedale* algae at JSC “Alvo”, Panevezys, Lithuania) was purchased from a local market (JSC “Hyper Maxima”, Kaunas, Lithuania). Refined sunflower oil (producer Ltd. “Floriol”, Budapest, Hungary) was obtained from a local market (JSC “Hyper Maxima”, Kaunas, Lithuania).

Calcium chloride (CaCl_2_, producer “Enolandia”, Fidenza, Italy) was purchased from the local market (JSC “Medeja”, Plunge, Lithuania). Organic pure lemon juice (producer “Alce Nero”, Bologna, Italy) was purchased from a “Livinn” company (Kaunas, Lithuania) (28 kcal, fat 0.1 g, carbohydrates 6.5 g (1.8 sugars), fibers 0.4 g and proteins 0.4 g).

Raw cow’s milk (pH 6.5, milk solids-not-fat 7.95%, total microorganisms < 5 × 10^4^ colony-forming units (CFU/mL), somatic cells < 7 × 10^5^ cells/mL, protein content 3.1%, fat content 3.5% and total solids 12.7%) was purchased from JSC “Rimi Baltic” (Kaunas, Lithuania). Prior to the experiments, raw milk was kept (for not longer than 1 h) in a refrigerator at +4 °C.

The general experimental design of this study is given in Figure 1. During the first stage of experiment (I), antimicrobial properties of the BIBs were tested, and the effect of the immobilization technology on BIBs acidity parameters (pH and total titratable acidity (TTA)), antioxidant properties (total phenolic compounds (TPC) content and 2,2-diphenyl-1-picrylhydrazyl (DPPH)-radical scavenging activity) and color characteristics was analyzed. During the second stage (II), non-immobilized (_NI_) and agar-immobilized (_AI_) BIBs were used for U-CC preparation, and their influence on U-CC sensory properties and induced emotions, acidity parameters, VC and FA profiles, BA concentration, color coordinates, moisture content and microbiological characteristics during the storage was evaluated.

### 2.2. Berry Industry By-Product (BIB) Immobilization and Unripened Cow Milk Curd Cheese Preparation

#### 2.2.1. Preparation of Agar-Immobilized (_AI_) Berry Industry By-Products (BIBs)

The agar powder was soaked in water (at 25 ± 2 °C for 1 h; 1 g agar powder in 20 mL of water) and further melted by heating for 5 min. Lyophilized raspberry, blueberry and elderberry BIBs were poured into the agar–water mixture (1 part of BIBs and 4 parts of agar–water mixture), and, with a syringe, drops of agar–water–BIB mixture were put into cold (+4 °C) refined sunflower oil. After obtaining a hard texture of the drop’s formation, they were removed from the oil, washed under a stream of warm water (+25 °C), dried at room temperature (+23 °C) for 12 h and, finally, used in further analyses and U-CC preparation. Images of the non-immobilized and immobilized BIBs are shown in Figure 2.

#### 2.2.2. Preparation of Unripened Cow Milk Curd Cheese (U-CC)

The U-CC was prepared using 18 L of raw cow milk per treatment. Raw cow milk intended for U-CC was pasteurized at 72–73 °C for 15–20 s, followed by cooling to 30 ± 2 °C. Then, 1% (*w*/*v*) of non-immobilized and 4% (*w*/*v*) of immobilized BIBs were added. For coagulation, pure organic lemon juice (30 mL per liter of milk) was used, in addition to the 0.2 g/L of milk CaCl_2_ that was added. After milk coagulation, the curd was mixed and gently cut into 200 g cubes and drained. Furthermore, the mass was placed into nylon containers and pressed (0.4 kg weight) for 12 h at 4 °C. U-CC samples without BIBs were analyzed as a control. In total, 7 groups of U-CC were prepared: (I) control—U-CC without BIB addition (C); (II, III and IV)—U-CC with non-immobilized raspberry, blueberry and elderberry BIB, respectively (C-Ras_NI_, C-Blu_NI_ and C-Eld_NI_, respectively) and (V, VI, VII)—U-CC with immobilized raspberry, blueberry and elderberry BIBs, respectively (C-Ras_AI_, C-Blu_AI_ and C-Eld_AI_, respectively). The U-CCs for chemical analysis were collected after 24 h (1 day) of the manufacturing process. For microbiological analyses, samples were collected after 1, 3, 4, 7 and 10 days of storage. Images of the U-CC are shown in Figure 3.

### 2.3. Methods for Berry Industry By-Product (BIB) Analyses

#### 2.3.1. Evaluation of the Antimicrobial Properties in Berry Industry By-Products (BIBs)

The antimicrobial activities of BIBs were evaluated against *Salmonella enterica Infantis* (*Salmonella enterica* subsp. *enterica serovar Infantis*), *Staphylococcus aureus*, *Escherichia coli* (hemolytic), *Bacillus pseudomycoides*, *Aeromonas veronii*, *Cronobacter sakazakii*, *Hafnia alvei*, *Enterococcus durans*, *Kluyvera cryocrescens* and *Acinetobacter johnsonii*, which were obtained from the Lithuanian University of Health Sciences (Kaunas, Lithuania) collection. The antimicrobial activity of the BIBs was assessed by measuring the diameter of inhibition zones (DIZ, mm) in agar-well diffusion assays as described previously by Balouiri et al. [24]. Accordingly, a 0.5 McFarland unit density suspension of each pathogen was inoculated onto the surface of cooled Mueller–Hinton agar (Oxoid, Basingstoke, UK) using sterile cotton swabs. Wells of 6 mm in diameter were punched in the agar and filled with 50 µL of the tested BIBs. Before the experiment, BIBs were diluted with a sterile physiological solution (1 g of the BIB was diluted with 2 mL of the physiological solution). Results were given as the average mean and standard error (SE) of the DIZ obtained from three parallel experiments (replicates).

#### 2.3.2. Evaluation of pH, Acidity and Color Characteristics in Berry Industry By-Products (BIBs)

The pH of BIB was evaluated with a pH meter (Inolab 3, Hanna Instruments, Venet, Italy). The color coordinates were analyzed by using a “Chromameter CR-400” (Konica Minolta, Tokyo, Japan). The total titratable acidity (TTA, °N) was determined for a 10 g of BIBs homogenized with 90 mL of distilled water and expressed as milliliters of 0.1 mol/L NaOH needed for neutralization of the mixture.

#### 2.3.3. Determination of Total Phenolic Compounds (TPCs) and 2,2-Diphenyl-1-picrylhydrazyl (DPPH^−^)-Radical Scavenging Activity

The TPC content of the BIBs was determined by a spectrophotometric method described by Vaher et al. [25]. All procedures in detail are described in Appendix A. The TPC content was expressed as mg of gallic acid equivalent mL of solution (mg GAE/100 g (DM)) [25]. The ability of the BIB extract to scavenge DPPH^−^ free radicals was assessed using the method described by Zhu et al. [26] (Appendix A).

### 2.4. Methods for Unripened Cow Milk Curd Cheese (U-CC) Analyses

#### 2.4.1. Evaluation of pH, Acidity, Color Coordinates, Texture Hardness, Moisture Content and Antioxidant Characteristics in Unripened Cow Milk Curd Cheese (U-CC)

The pH of U-CC was evaluated with a pH meter (Inolab 3, Hanna Instruments, Venet, Italy) by inserting the pH meter electrode directly into the U-CC sample. The color coordinates were analyzed with a Chromameter CR-400 (Konica Minolta, Tokyo, Japan). For color coordinate evaluation, the U-CC was cut into slices, and the color indicators were immediately measured on the surface of the U-CC. For TTA analysis, a U-CC slurry was prepared by blending 20 g of grated U-CC with 12 mL of water. Then, a 20 g sample was mixed with 250 mL of distilled water and filtered through a Whatman #1 filter paper. Furthermore, 25 mL of the filtered sample was titrated with 0.1 mol/L NaOH, and phenolphthalein was used as an indicator. The TTA was expressed in Terner degrees (°T). The texture hardness was evaluated by using a texture analyzer (Brookfield, Ametek, Middleboro, Massachusetts, USA). For texture hardness evaluation, the U-CC was cut into 2 cm-thick slices. The moisture content was determined according to the ICC standard method 110/1 (1976) by drying the sample at 103 ± 2 °C until reaching constant weight [27].

The methods for determination of TPC content and DPPH^−^ radical scavenging activity are described above in Section 2.3.3.

#### 2.4.2. Evaluation of Fatty Acid (FA) Profile in Unripened Cow Milk Curd Cheese (U-CC)

The extraction of lipids for FA analysis was performed with chloroform/methanol (2:1 *v*/*v*), and fatty acid-methyl esters (FAME) were prepared according to the method described by Pérez Palacios et al. [28]. All procedures are described in detail in Appendix A.

#### 2.4.3. Evaluation of Volatile Compounds (VCs) in Unripened Cow Milk Curd Cheese (U-CC)

The VCs of U-CC were analyzed by gas chromatography–mass spectrometry (GC-MS) as described by Bartkiene et al. [29] with slight modifications described in detail in Appendix A.

#### 2.4.4. Evaluation of Biogenic Amine (BA) Content in Unripened Cow Milk Curd Cheese (U-CC)

The extraction and determination of BA in U-CC followed the procedures developed by Ben-Gigirey et al. [30], with some modifications as described by Bartkiene et al. [31], and are described in detail in Appendix A.

#### 2.4.5. Evaluation of Changes in Total Lactic Acid Bacteria (LAB), Total Bacteria (TBC) and Total Enterobacteria (TEC) Viable Counts in Unripened Cow Milk Curd Cheese (U-CC) during Storage

The U-CC samples were subject to microbiological analyses (total viable bacteria (TBC), total viable lactic acid bacteria (LAB) and total viable enterobacteria (TEC)). Microbiological analyses of the U-CC were performed after 1, 3, 4, 7 and 10 days of storage in a refrigerator at +4 °C.

For analysis, 10 g of U-CC was homogenized with 90 mL of saline (9 g/L NaCl solution). Serial dilutions of 10^4^–10^8^ with saline were used for the sample preparation. Evaluation of TBC on plate count agar (PCA, CM0325, Oxoid Ltd., Basingstoke, UK), LAB on The Man Ragosa Sharpe agar (MRS, CM0361, Oxoid Ltd., Basingstoke, UK) and TEC on MacConkey agar (CM0115, Oxoid Ltd., Basingstoke, UK) was undertaken. The number of viable microorganisms was counted in the dilutions containing between 30 and 300 colonies and expressed as log_10_ of colony-forming units per gram (CFUs/g) [32]. All results were expressed as the mean value of three determinations and standard error.

#### 2.4.6. Evaluation of Overall Acceptability and Induced Emotions for Consumers in Unripened Cow Milk Curd Cheese (U-CC)

The overall acceptability of the U-CC was established by 10 trained judges, according to the International Standards Organization (ISO) method 6658:2017 [33], using a 10-point scale ranging from 0 (“extremely dislike”) to 10 (“extremely like”). Ten judges were recruited internally (Institute of Animal Rearing Technologies and Department of Food Safety and Quality, Lithuanian University of Health Sciences, Kaunas, Lithuania): 5 females and 5 males, from 25 to 50 years old [34,35]. Individuals who were familiar with this study were excluded from the panel. The previous training of the judges was based on descriptive analysis [36,37,38]. Selected judges were non-smokers, interested in sensory analysis and motivated to participate.

In parallel, U-CC samples were tested by applying FaceReader 6.0 software (Noldus Information Technology, Wageningen, The Netherlands), scaling eight emotion patterns (neutral, happy, sad, angry, surprised, scared, disgusted and contempt) [39]. All procedures are described in detail in Appendix A.

### 2.5. Statistical Analysis

The results are expressed as the mean ± standard error (for BIB antimicrobial properties and physicochemical parameters, *n* = 3; for U-CC physicochemical and microbiological parameters, *n* = 3; for U-CC overall acceptability and emotions induced for consumers, *n* = 10). The data were analyzed using the statistical package SPSS for Windows (v27.0, SPSS Inc., Chicago, IL, USA). The normal distribution of data was checked using Descriptive Statistics tests. In order to evaluate the influence of the different types of BIBs and immobilization on the analyzed U-CC parameters, data were evaluated by the multivariate analysis of variance (ANOVA) procedure and Tukey’s honestly significant difference (HSD) procedure as post hoc tests. A linear Pearson’s correlation was used to quantify the strength of the relationship between the variables (0.00–0.19, very weak; 0.20–0.39, weak; 0.40–0.59, moderate; 0.60–0.79, strong and 0.80–1.0, very strong) [40]. Results were recognized as statistically significant at *p* ≤ 0.05.

## 3. Results and Discussion

### 3.1. Characteristics of the Non-Immobilized (_NI_) and Immobilized (_AI_) Berry Industry By-Products (BIBs)

#### 3.1.1. Antimicrobial Properties of Non-Immobilized (_NI_) Berry Industry By-Products (BIBs)

The diameters of inhibition zones of the BIBs against the tested pathogens are shown in Table 1. All the tested BIBs showed antimicrobial activity against *Enterococcus durans*. However, antimicrobial activity of the tested BIBs against *Salmonella enterica Infantis* and *Kluyvera cryocrescens* was not established. The broadest spectrum of pathogen inhibition was shown by raspberry BIBs (inhibited 7 out of 10 tested pathogens), and the highest DIZ against *Bacillus pseudomycoides* was found (15.5 mm). Blueberry BIBs showed antimicrobial properties against 5 out of 10 tested pathogens. The lowest spectrum of pathogen inhibition was found in elderberry BIBs (inhibited 2 out of 10 tested pathogens).

It was reported that berries possess a wide range of activities, including antioxidant and antibacterial ones because of the presence of flavonoids, phenolic acids and tannins [41]. It was reported that raspberry phenolic extracts inhibit non-virulent *Salmonella* [42]. The study of Nohynek et al. [43] demonstrated that raspberry extract inhibits *Helicobacter pylori*, *Bacillus cereus*, *Staphylococcus aureus*, *Staphylococcus epidermidis*, *Campylobacter jejuni* and *Clostridium perfringens*. Raspberry extract antibacterial activity was explained by the presence of phenolic compounds, from which ellagitannin fraction was pointed out as the most important [43]. In addition to extracts, Puupponen-Pimiä et al. [42] stated that *Typhimurium* spp. and *Staphylococcus aureus* were suppressed by lyophilized raspberry.

Blueberries contain anthocyanins, which exhibited inhibition of the proliferation of many pathogens, including *E. coli* and *S. aureus* [44]. Chlorogenic acid, quercetin, ellagic acid and quercetin-3-galactoside were indicated to be the main antimicrobial compounds in blueberry extract [45].

Due to their phytochemicals activities (especially anthocyanins), elderberries possess antiviral and anti-inflammatory properties [46]. Vatai et al. reported [47] that the anthocyanin content in elderberries is much higher than in grapes. However, elderberries showed a higher pH value, in comparison with other berries, and this characteristic can lead to lower antimicrobial activity in this kind of berry.

Finally, antimicrobial activity of berries depends on the synergy of various compounds. The most important are organic acids, phenolic acids, tannins and their combinations [43]. It should be pointed out that also the effect of pH is very important for berry antimicrobial characteristics. Organic acids are membrane-active constituents, which damage the inner cell membrane in their undissociated form [41]. Additionally, acids alter the membrane permeability of the microbial cell and acidify the cytoplasm [48]. However, Ördögh et al. [48] reported that the antibacterial activity of juice and pomace extracts is independent on pH values and that non-dissociable compounds are responsible for the pathogens growth inhibition. More research is needed to identify the antimicrobial mechanisms of such a complex matrix. Nevertheless, this study showed that the tested BIBs possess desirable antimicrobial activities and could be very prospective ingredients for U-CC enrichment.

#### 3.1.2. Antioxidant Characteristics, Color Coordinates (L*, a* and b*), pH and Acidity (TTA) Parameters of Berry Industrial By-Products (BIBs)

In comparison to BIB antioxidant properties, the highest DPPH^−^ radical scavenging activity was obtained in non-immobilized raspberry and elderberry BIBs (on average, 76.5%) (Table 2). However, comparing TPC content, the highest TPC content was displayed by non-immobilized elderberry BIBs (143.6 mg GAE/100 g). A very strong positive correlation between BIB DPPH^−^ radical scavenging activity and TPC content was established (r = 0.936, *p* ≤ 0.001). Multivariate analysis of variance showed that the variety of berry was a significant factor in both DPPH^−^ radical scavenging activity and TPC content in BIBs (*p* = 0.028 and *p* = 0.017, respectively), although immobilization and immobilization–berry variety interaction were significant, just on DPPH^−^ radical scavenging activity of the BIBs (*p* = 0.008 and *p* = 0.026, respectively).

Essentially, the BIBs, i.e., those obtained after juice manufacturing, consist of pulp, peels and seeds [49] and contain considerable amounts of anthocyanins, especially in the dark color berries. It was reported that raspberry, blueberry and elderberry BIBs are rich in these compounds and possess high antioxidant activity [50,51,52,53,54,55]. Notably, the DPPH radical scavenging activity depends on the berry variety, region of cultivation and the chosen extraction and drying method. Četojević-Simin et al. reported that raspberry cultivar ‘Willamette’ BIB extracts possess 43.7 ± 2.02 mg GAE/g of total phenolic content [56]. Our results showed that the highest TPC content is obtained in elderberry BIBs. Tánska et al. reported that the TPC content in elderberry pomace is 13.86 ± 0.22 g/100 g [57]. In comparison with blueberries, elderberries have higher anthocyanin and phenolic compound contents [58]. Moreover, antioxidant activity can be a result of synergic interactions among antioxidant compounds [59].

In comparison to BIB color coordinates, the highest lightness (L*) was attained in non-immobilized and immobilized raspberry BIBs (on average, 32.6 NBS), and immobilization was not a significant factor on BIB L*. However, a negative strong correlation between BIB L* and pH was established (r = −0.639 and *p* = 0.004). Opposite to L*, raspberry BIBs showed the highest redness coordinate (a*) (non-immobilized 29.6 NBS and immobilized 20.5 NBS). Also, a negative strong correlation between BIB a* and pH values was found (r = −0.821 and *p* ≤ 0.001). Both analyzed factors (berry variety and immobilization) and their interaction were significant in a* values of the BIBs (*p* ≤ 0.001, *p* = 0.028, and *p* = 0.035, respectively). Similar tendencies with the yellowness coordinate (b*) were obtained; viz. in comparison with non-immobilized raspberry BIBs, non-immobilized blueberry and elderberry BIBs showed, on average, 3.59 and 3.15 times lower b* values, respectively, and, in comparison with immobilized raspberry BIBs, immobilized blueberry and elderberry BIBs showed, on average, 26.3 and 5.0 times lower b* values, respectively. A negative moderate correlation between BIB b* and pH values was found (r = −0.716 and *p* ≤ 0.001).

BIBs have a wide impact on food color and taste formation and can form unique aromas [60]. The color of food improves not only its aesthetic value but also exerts influence on consumer’s behavior. Therefore, food enrichment with berries can improve the end-product (e.g., cheese) color and taste and boost the appeal to consumers, who are searching for more healthy, attractive and functional food [61].

Berry anthocyanins are the main compounds responsible for color intensity, and they can remain stable in high acidity food products [62]. It was conveyed that the most abundant pigments in red raspberry fruits are cyanidin-3-sophoroside, cyanidin-3-(2^g^-glucosylrutinoside), cyanidin-3-glucoside and cyanidin-3-rutinoside [63], and a comparable composition of anthocyanins in red raspberry fruits has been reported by other authors [64,65,66]. Blueberries contain different types of anthocyanins [67], including malvidin, delphinidin, petunidin, cyanidin and peonidin [68]. The most abundant anthocyanins in elderberries are cyanidin-3-O-sambubiozides, cyanidin-3-sambubiozides-5-glucosides, cyanidin-3-O-glucoside and cyanidin-3,5-diglucoside [69].

The lowest pH was attained in non-immobilized raspberry BIBs (3.29). However, the highest TTA was found by immobilized elderberry BIBs (1.00 °N) (Table 2). Expectedly, a correlation between BIB pH and TTA values was not found. Berry variety and immobilization were significant factors on BIB pH (*p* = 0.14 and *p* = 0.35, respectively). It was reported that the TTA of European red raspberries is, on average, 4.90, and the pH is, on average, 3.19 [70]. Sargent et al. testified that blueberry pH can vary from 3.36 to 3.62, that TTA can vary from 0.33 to 0.62 and that acidity parameters depend on harvest maturity [71]. The predominant organic acids in blueberries are quinic and citric acids, and also fresh blueberries contain a broader variety of organic acids, such as malic, oxalic and fumaric acids [72]. Citric acid in blueberries could range from 1.86 ± 0.01 to 13.42 ± 0.38 mg 100/g [72]. In elderberries, the predominant organic acids are citric, malic, shikimic and fumaric acids [73], and the elderberry TTA could vary from 0.52 to 0.94 g 100/g [73], and the pH can vary from 4.58 to 5.45.

### 3.2. Characteristics of the Unripened Cow Milk Curd Cheese (U-CC)

#### 3.2.1. pH and Acidity (TTA) Parameters, Color (L*, a* and b*) Parameters, Texture Parameters and Antioxidant Characteristics of Unripened Cow Milk Curd Cheese (U-CC)

The pH and acidity (TTA) parameters, color characteristics and texture hardness of U-CC are tabulated in Table 3. Observing U-CC acidity parameters, the highest pH and the lowest TTA are shown in the control U-CC samples (5.80 and 1.90 °T, respectively). Despite that, significant differences between TTA of U-CC samples, such behavior, were not found when they were prepared with BIBs. The lowest pH values were obtained in U-CC with raspberry BIBs (significant differences between immobilized and non-immobilized U-CC pH were not found, and pH was, on average, 5.29). Immobilization was not a significant factor in U-CC pH values, and U-CCs with blueberry BIBs had a pH of, on average, 5.37, and, in turn, U-CCs with elderberry BIBs had a pH of, on average, 5.60.

Acidity influences the final flavor of cheeses, as well as the biochemical, textural and functional properties [74]. In our study, lemon juice was used for U-CC preparation, which is a commonly used ingredient in the U-CC industry to reduce manufacturing time [75]. The main (more than 90%) organic acid (on average, 73.94 g/L) in lemon juice is citric acid [76]. Also, citric and malic acids are the main organic acids in most of the berry fruit species [77,78,79,80,81]. Finally, it can be stated that the addition of BIBs was an additional source of organic acids in U-CC, which led to lower U-CC pH and higher TTA values.

Several factors influence the color of anthocyanins, including pH and degree of hydroxylation [82]. In comparison to U-CC color characteristics, the highest lightness (L*) was reached with the control U-CC (99.5 NBS). Other U-CC sample L* values were lower (on average, by 29.2% for C-Eld_AI_, and, on average, by 9.67% for C-Ra_NI_ and C-Ra_AI_), in comparison with control samples. In comparison sample a* (redness or -a* greenness) coordinates, the highest greenness was observed in the control samples (−4.68 NBS). Also, U-CC samples prepared with immobilized raspberry and blueberry BIBs showed negative a* coordinates (−2.71 and 1.94 NBS, respectively). Other U-CC samples showed positive a* coordinates, which ranged from 1.07 NBS (U-CC samples prepared with non-immobilized raspberry BIB) to 7.35 NBS (U-CC samples prepared with non-immobilized elderberry BIB). The highest yellowness (b*) was found in the control U-CC samples (25.0 NBS), and the lowest b* values were found in C-Blu_NI_, C-Eld_NI_ and C-Eld_AI_ samples (on average, 7.81 NBS). Tests of between-subject effects showed that the type of BIB and immobilization were significant factors on U-CC a* and b* coordinates (on a* coordinates *p* < 0.001 and *p* = 0.004, respectively; on b* coordinates *p* < 0.001 and *p* = 0.010, respectively), and the interaction of these factors was significant upon U-CC a* coordinates (*p* < 0.001). Similar to our results, Guiné et al. reported reduced lightness and yellowness as well as an increased redness in fresh cheese enriched with red fruits (fresh raspberry, fresh blueberry, frozen blueberry and a mixture of fresh raspberry and blueberry) [83].

In all cases, BIBs increased the hardness of the U-CC samples, and the hardest structure was found in U-CC samples prepared with immobilized raspberry and elderberry BIBs (0.500 mJ). Comparing U-CC samples prepared with non-immobilized and immobilized BIBs, one found that in all cases a harder structure was detected in U-CC samples prepared with immobilized BIBs. Berries are a good source of phenolic acids, which possess antioxidant properties, and their typical representatives encompass hydroxybenzoic acids (e.g., gallic, *p*-hydroxybenzoic, vanillic and syringic acids) and hydroxycinnamic acids (e.g., ferulic, caffeic, *p*-coumaric, chlorogenic and sinapic acids) [82,84]. It was reported that the phenolic acids are added to some dairy products as functional ingredients [85]. However, phenolic acids can interact with various types of dairy proteins and form phenolic acid–protein complexes [86,87,88,89,90]. These interactions can change protein structure and affect the physicochemical attributes of the system, including protein solubility and emulsification, among other properties [86,91,92]. The study of Masmoudi et al. [93] revealed that fortification with *A. unedo* fruit extract also increased the firmness of soft “Sardaigne” cheese.

Our results showed that despite significant differences in U-CC moisture content not being established, positive moderate and strong correlations were found between U-CC texture hardness and TPC content (r = 0.575 and *p* = 0.006) as well as with DPPH^−^ radical scavenging activity (r = 0.821 and *p* < 0.001). These findings can be explained by possible interaction of BIB antioxidant compounds and U-CC proteins. However, further studies are needed to explain exactly the mechanism of such interactions.

The TPC content and DPPH^−^ radical scavenging activity of U-CC samples are given in Figure 4. In all cases, non-immobilized and immobilized BIB addition positively increased TPC content and DPPH^−^ radical scavenging activity of the U-CC, in comparison with control U-CC samples. The highest TPC content was found in the C-Ra_NI_ sample group (184.5 ± 6.82 mg 100/g). In comparison with C-Ra_NI_, C samples showed, on average, 74.2% lower TPC content. C-Blu_NI_, C-Ra_AI_, C-Blu_AI_ and C-Eld_AI_ samples showed, on average, 34.0% lower TPC content. C-Eld_NI_ samples showed, on average, 14.2% lower TPC content. Comparing TPC content of U-CC prepared with immobilized and non-immobilized BIBs, immobilization did not have a significant effect on TPC content in U-CCs prepared with blueberry BIBs (in C-Blu_NI_ and C-Blu_AI_ samples, TPC content was, on average, 123.8%). However, U-CCs prepared with immobilized raspberry and elderberry BIBs showed, on average, 32.9 and 26.0% lower TPC content, respectively, in comparison with U-CCs prepared with non-immobilized raspberry and elderberry BIBs. As expected, a strong positive correlation between TPC content and DPPH^−^ radical scavenging activity of the U-CC was established (r = 0.658 and *p* = 0.001).

It was stated that antioxidant activity of the fruits degrades at a higher rate than total phenols and other antioxidant compounds, including ascorbic acid, and these findings corroborate the fact that antioxidant activity is influenced cumulatively by many factors, including TPC content [94]. Likewise, taking into consideration that phenolic compounds can interact with various types of dairy proteins, their interactions with agar molecules might also be possible. Despite agarose being the idealized structure of agar [95]—which consists of repeating units of agarobiose or LA-G_n_ [96], alternating with β-d-galactopyranosyl and 3,6-anhydro-α-l-galactopyranosyl groups [97]—other groups such as sulphate esters, methyl ethers or pyruvate acid ketals are also present in the agar structure [98]. To the best of our knowledge, unripened curd cheese with berry by-products such as those included in our study has not been tested by other researchers. However, Gonçalves et al. [99] reported that the addition of red fruits (blueberry and raspberry) in fresh cheeses increased the levels of phenolic compounds and improved antioxidant activity. Lucera et al. [100] enriched spreadable cheese with flours from red and white grape pomace, tomato peel, broccoli, corn bran and artichokes. He found that total phenolic content and antioxidant activity significantly increased in enriched samples. Similar tendencies were observed in the study of Masmoudi et al. [93], where *A. unedo* fruit extract inclusion in a soft “Sardaigne” cheese improved its DPPH scavenging activity.

Comparing DPPH^−^ radical scavenging activity of the U-CC prepared with immobilized and non-immobilized BIBs, one found out that, in all cases, samples prepared with immobilized BIBs showed lower DPPH^−^ radical scavenging activity, in comparison with samples prepared with non-immobilized ones (U-CC prepared with immobilized raspberry, blueberry and elderberry was, on average, 10.2, 10.0 and 10.2% lower, respectively). However, tests of between-subjects effects showed that analyzed factors (type of BIB and immobilization) and their interaction were not statistically significant on TPC content and DPPH^−^ radical scavenging activity of U-CC.

#### 3.2.2. Fatty Acid (FA) Profile of Unripened Cow Milk Curd Cheese (U-CC)

The fatty acid (FA) profile of U-CC samples and the influence of the analyzed factors (type of BIBs and immobilization) and their interaction on FA profile are shown in Table 4. The main FA in U-CC was palmitic acid, and its content in U-CC ranged from 21.5 to 33.8% from the total fat content (in samples C-Eld_AI_ and sample groups C and C-Eld_NI_, respectively). The type of BIBs, immobilization and these factor’s interaction were significant on palmitic acid content in U-CC (*p* < 0.001, *p* = 0.007 and *p* < 0.001, respectively). Other predominant Fas in U-CC were myristic, stearic and oleic acids. The type of BIB was a significant factor on myristic acid content in U-CC (*p* = 0.028). The lowest butyric, caproic, capric, lauric, myristoleic, margaric and palmitoleic acid content was found in C-Eld_AI_ samples (2.05, 1.21, 1.68, 1.77, 0.520, 0.220 and 1.06% from the total fat content, respectively). The type of BIB and analyzed factor interaction were statistically significant on butyric (*p* = 0.018 and *p* < 0.001, respectively) and palmitoleic (*p* = 0.003 and *p* < 0.001, respectively) acids in U-CC, and the analyzed factor interaction was also significant on capric acid in U-CC (*p* = 0.035). The highest caprylic acid content was observed in C-Ra_AI_ samples (1.13% from the total fat content). The highest pentadecylic acid content was found in C, C-Eld_NI_ and C-Blu_AI_ sample groups (on average, 1.07% from the total fat content); the highest linoleic acid content was obtained in C-Ra_NI_ and C-Eld_AI_ samples (on average, 11.0% from the total fat content); and the highest α-linolenic acid content was established in C-Blu_NI_ samples (on average, 8.04% from the total fat content). Moreover, analyzed factor interaction was significant for the FA content in U-CC (*p* = 0.025, *p* = 0.044 and *p* = 0.029, respectively). Arachidic acid was found in three out of seven U-CC samples (C-Blu_NI_, C-Blu_AI_ and C-Eld_AI_); furthermore, gondoic acid was not established in C and C-Ra_NI_ samples. Analyzed factors and their interaction were significant in arachidic FA content in U-CC samples (*p* < 0.001, *p* = 0.011 and *p* < 0.001, respectively). The predominant FA group in U-CC samples was saturated fatty acids (SFAs), where the lowest content of SFAs was found in C-Eld_AI_ samples (in comparison with C samples, on average, by 1.6 times lower). The type of BIBs and analyzed factor interaction were significant on SFA content in U-CC (*p* = 0.035 and *p* = 0.006, respectively). The highest monounsaturated (MUFA) and polyunsaturated (PUFA) fatty acid contents were attained in C-Eld_AI_ samples (40.0 and 16.2% from the total fat content), whereas other U-CC samples showed values from 1.65 (C-Ra_AI_, C-Ra_NI_, and C-Eld_NI_) to 1.41 (C-Blu_AI_) times lower MUFA content and from 5.89 (C-Eld_NI_) to 1.05 (C-Ra_NI_) times lower PUFA content, in comparison with C-Eld_AI_ samples. The analyzed factor interaction was statistically significant for MUFA content in U-CC (*p* = 0.019), and both analyzed factors and their interaction were significant on PUFA content in U-CC (*p* < 0.001). Contrasting omega-3, -6 and -9 contents, the predominant FA group was omega-9, with the highest content in C-Eld_AI_ samples (38.4% from the total fat content), and both analyzed factors and their interaction were significant on omega-9 content in U-CC (*p* < 0.001). The highest content of omega-6 was reached in C-Ra_NI_ and C-Eld_AI_ samples (on average, 11.0% from the total fat content), and the analyzed factor interaction was significant for omega-6 content in U-CC (*p* = 0.028). Omega-3 fatty acid content in U-CC samples ranged from 0.630 to 8.04% of the total fat content, in C-Eld_NI_ and C-Blu_NI_ samples, respectively.

A typical cow’s milk FA profile consists, on average, of 70% SFA, 25% MUFA and 5% PUFA [101]. However, the FA profile of cheese varies, which is related to the milk origin and animal rearing conditions, and it also depends on the cheese-making technology [102]. In the human diet, dairy products are important sources of SFA (especially, C12:0, C14:0 and C16:0) as well as other Fas that have a beneficial effect on health (butyric acid, branched FA, odd FA, oleic FA and conjugated linoleic fatty acid *cis*9*trans*11 C18:2) [103]. However, consuming dairy products that contain high content of SFA is related to the development of many diseases [102]. In addition, different Fas—each of which have a specific physiological function and may affect lipoprotein metabolism in a different way—can carry the risk of contributing to many diseases [104,105,106]. It was reported that butyric acid shows anti-inflammatory [107,108] effects, and branched-chain Fas elicit anti-carcinogenic effects [109,110]. Available research data state that the average MUFA content in various cheeses is 179.90 mg/g of fat, and the average PUFA content is at a similar level [102,111]. Dietary *n*-3 polyunsaturated fatty acids are recommended for heart disease prevention, and linolenic acid exhibits anti-carcinogenic and anti-atherogenic effects [112,113,114], whereas *n*-6 PUFAs improve sensitivity to insulin [115]. However, taking into consideration that dairy products are rich in SFA, their enrichment with BIBs becomes very prospective. It was reported that raspberry consumption may reduce vascular oxidative stress induced by a high-fat and high-sucrose diet via reducing nitric oxide oxidation despite increasing nitric oxide synthase in vivo [116]. Additionally, the fatty acid profile of blueberry seed oil contains more than 65% PUFA [117]. Also, raspberry seed oil contains more than 86.2% of PUFA [118], and α-linolenic acid consists of about 50% of the FA profile in blueberry seed oil, followed by linoleic acid [119]. Elderberry seeds contain 22.4 g 100/g lipids (from the seed dry weight) [120], and nineteen Fas were identified in elderberry. It was then concluded that these berries are a balanced source of essential PUFAs for human health, because of their high levels of α-linolenic acid [18]. Finally, BIBs are very valuable ingredients for the food industry, as they contain all biological parts of the berry fruit (seeds and outer layer), where the main bioactive compounds, including Fas, are concentrated.

#### 3.2.3. Volatile Compound (VC) Profile of Unripened Cow Milk Curd Cheese (U-CC)

The VC profiles of U-CC are presented in Table 5. In total, 43 VCs were identified in U-CC samples, chiefly 5 aldehydes, 6 ketones, 19 terpenoids, 4 organic acids and 6 aliphatic hydrocarbons, as well as 3,4-dimethylbenzyl alcohol, 1,3-bis(1,1-dimethylethyl)benzene and decanoic acid, ethyl ester (Table 5). The main VC in all U-CC sample groups was D-limonene, and its content in U-CC samples ranged from 36.2% for the total VC content (in sample C-Ra_NI_) to, on average, 57.6% for the total VC content (in samples C, C-Eld_NI_, C-Ra_AI_, C-Blu_AI_ and C-Eld_AI_). Both analyzed factors (type of BIB and immobilization) and their interaction were statistically significant on D-limonene content in U-CC (Table 6). This finding can be explained by technological features, because lemon juice was used for U-CC production, which could be a source of D-limonene [121]. Other VCs identified in all U-CC samples and for which the content was higher than 1% for the total VC content are 2-heptanone, 2-nonanone, β-pinene, γ-terpinene, hexanoic acid, octanoic acid and decanoic acid. 2-Heptanone has a soap odor, and 2-nonanone odor is described as having a hot milk and soap odor. These VCs were reported as key aroma compounds in Cheddar cheese [122]. A test of between-subject effects showed that immobilization was a significant factor on 2-heptanone, 2-nonanone and octanoic acid content in U-CC samples (Table 6). Β-Pinene has a cooling, woody, piney and turpentine odor, with traces of fresh mint, eucalyptus and camphor-like aroma [123]. Γ-Terpinene is a typical compound of several essential oils such as citrus, savory and oregano, among others [124]. Hexanoic acid odor is described as sweaty, whereas octanoic acid odor is described as burnt waxy, body odor and sweat [122], and decanoic acid odor is described as fatty, soapy, dust, waxy and burned [125]. Both analyzed factors (type of BIB and immobilization) and their interaction were significant on β-pinene, hexanoic acid and decanoic acid content in U-CC samples (Table 6). The higher contents of 2-nonanone, nonanal, benzaldehyde and decanoic acid in the U-CC samples could be related to the presence of elderberry by-products because these are the main VC of these berries [126]. Raspberry by-products may enhance the levels of key VCs such as hexanal, benzaldehyde, β-pinene, β-myrcene, sabinene, D-limonene and hexanoic acid in the U-CC samples [127]. The increase in hexanal, α-pinene, β-pinene, β-myrcene, α-terpineol and hexanoic acid could be related to the addition of blueberry by-products in the U-CC samples [127].

In all cases, U-CC samples prepared with BIBs showed a higher variety of VCs, in comparison with the control U-CC (the number of VCs identified were 26 in C samples, 31 in C-Ra_NI_, 27 in C-Blu_NI_, 29 in C-Eld_NI_, 32 in C-Ra_AI_, 27 in C-Blu_AI_ and 30 in C-Eld_AI_). This tendency was also observed by Pluta-Kubica et al. [128] who examined the volatile profile of fresh soft rennet-curd cow milk cheese with wild garlic leaves. To the best of our knowledge, unripened cow milk curd cheese with berry by-products has not previously been investigated for a VC profile.

#### 3.2.4. Biogenic Amine (BA) Content of Unripened Cow Milk Curd Cheese (U-CC)

Biogenic amine concentrations in U-CC are presented in Table 7. In the control U-CC, Bas were not established. Tryptamine, cadaverine, histamine, tyramine and spermine (0.870, 19.9, 8.91, 11.2 and 21.0 mg/kg, respectively) were found just in the C-Eld_AI_ sample group. Putrescine was established in two out of seven U-CC samples (C-Eld_NI_ and C-Eld_AI_). Spermidine was formed in four out of seven U-CC samples (in C-Ra_NI_, C-Blu_NI_, C-Eld_NI_ and C-Eld_AI_ samples, spermidine content was 2.40, 11.6, 1.40 and 19.7 mg/kg, respectively). A moderate positive correlation between spermidine concentration and pH values was found (r = 0.523 and *p* = 0.015), and the type of BIB and BIB*immobilization interaction was significant for this BA content in U-CC (*p* = 0.031 and *p* < 0.001, respectively). Bas can be easily formed in fermented dairy product by decarboxylation of amino acids [129]. The most common Bas in dairy products are histamine, tyramine, putrescine and cadaverine [130]. Taking into consideration that the individual toxicological threshold for BA concentration can vary from a few mg/kg (in a sensitive person) to approximately 100 mg/kg (in a healthy person) [131], it is very important to control BA in the end products, to ensure as much as possible a lower concentration of these compounds. The main mechanism of BA occurrence in dairy products is decarboxylation activity of LAB strains, as well as undesired bacteria [130,132]. Synthesis of Bas is a multifactorial process which depends on many factors (initial contamination, starter cultures, maturation and storage time), including environmental conditions [129]. It was reported that the highest amounts of BA can be found in the fermented or ripened dairy products [133]. In our study, the obtained results (low content of Bas) can be explained by the employed U-CC technology, which does not include the fermentation process. However, data about BA in unripened cheeses are scarce. Ercan et al. [134] reported that cadaverine, histamine, phenylethylamine, tyramine, tryptamine, putrescine and spermidine were found in fresh Turkish cheese kashar, but BA content was significantly lower compared to mature kashar.

#### 3.2.5. Microbiological Parameters of Unripened Cow Milk Curd Cheese (U-CC) during Storage

Changes in LAB, TBC and TEC in U-CC after 1, 3, 4, 7 and 10 days of storage are depicted in Figure 5. After 1 day of storage, the growth of enterobacteria in U-CC was not established, and LAB viable counts in U-CC were, on average, 5.20 log_10_ CFUs/g. Also, significant differences between TBC in U-CC samples were found. In C, C-Ra_AI_ and C-Blu_AI_ sample groups, TBC was, on average, 2.35 log_10_ CFU/g, and in C-Ra_NI_, C-Blu_NI_, C-Eld_NI_ and C-Eld_AI_ sample groups, TBC was, on average, 2.53 log_10_ CFUs/g. After 2 days of storage, LAB viable counts in U-CC were, on average, 5.08 log_10_ CFUs/g; TBC was, on average, 2.72 log_10_ CFUs/g; and TECs were not established. After 3 days of storage, significant differences on TBC and LAB viable counts in most of the U-CC samples were not detected. However, after 4 days of storage, the highest TBC was found in the C-Blu_NI_ sample group (on average, 3.22 log_10_ CFUs/g), and a trend was observed throughout storage time, chiefly in which TBC in U-CC increased and LAB viable counts decreased. After 4 days of storage, TECs were established in three out of seven U-CC samples (in C, C-Blu_NI_ and C-Eld_NI_), and, after 10 days of storage, TECs were found in all U-CC samples. Additionally, the highest TEC number was found in the C-Blu_NI_ sample group (on average, 1.88 log_10_ CFU/g). Moderate positive correlations were found between TBC in U-CC samples after 1 day of storage and spermidine content in U-CC (r = 0.545 and *p* = 0.011). However, no correlations were detected between microbiological and acidity parameters, as well as with U-CC antioxidant characteristics.

The data about microbiological activity In unripened cheese are limited. Sturza et al. [135] reported that berry (rose-hip, aronia, sea buckthorn and hawthorn) powders induce the reduction of pathogenic microorganisms in cream cheese. Manipulating product formulation has long been a method to prevent growth of spoilage microorganisms in dairy products, especially by using chemical preservatives [136]. However, consumers are looking for more natural alternatives. Our previous studies showed that berry and fruit industry by-products possess desirable antimicrobial properties and are prospective ingredients for traditional food formula enrichment [137,138]. As the dairy industry is interested in reducing waste, it is imperative that much attention is paid to microbial spoilage reduction. Using natural approaches, described in this study, the dairy industry may adopt sustainable strategies to improve the functional value and reduce microbial spoilage of U-CC. Importantly, the suggested technology can be effective for both stakeholders (the dairy and berries industries), because it not only reduces U-CC spoilage but also increases BIB effective valorization.

#### 3.2.6. Overall Acceptability and Induced Emotions for the Judges of Unripened Cow Milk Curd Cheese (U-CC)

Significant differences between the overall acceptability of U-CCs were not obtained, and, on average, the overall acceptability of the U-CC was 8.34 points. Nevertheless, significant differences were found between the emotional intensity induced in the judges by the U-CC (Figure 6). The predominant emotion expressed by judges during the U-CC testing was “neutral”. The “neutral” emotion intensity ranged from 0.564 (C-Ra_NI_ and C-Blu_NI_ samples) to 0.736 (C samples) (Figure 6a). The domination of the “neutral” emotion can be explained by the traditional taste of this product. Indeed, most of the judges were familiar with such a type of tested dairy product. In comparison of the expression intensity of the emotion “happy”, the highest intensity of “happy” expression was induced by C-Eld_NI_ samples (0.137). In other samples, the induced intensity of “happy” emotional expression was 35.8 to 84.7% lower (C-Eld_AI_ and C-Rad_NI_ samples, respectively) (Figure 6b). Despite the lowest intensity of the “sad” emotion being induced by testing C-Eld_AI_ samples (Figure 6c), the same sample group induced the highest intensity of the “angry” emotion (Figure 6d). The intensity of the emotion “surprised” ranged, on average, from 0.039 (induced by sample groups C, C-Blu_NI_, C-Eld_NI_ and C-Ra_AI_) to 0.012 (induced by sample groups C-Ra_NI_, C-Blu_AI_ and C-Eld_AI_) (Figure 6e). Comparing the intensity of the emotion “scared” for the judges, samples C-Ra_NI_, C-Eld_NI_, C-Ra_AI_ and C-Blu_AI_ was induced, on average, with two times higher intensity than the control ones (Figure 6f). The highest intensity of the emotion “disgusted” was detected when C-Ra_NI_ and C-Blu_nNI_ samples were tested, and slightly lower (however, significant not different) intensity of this emotion was induced by C and C-Eld_AI_ sample groups (Figure 6g). The lowest intensity of the emotion “contempt” for judges was induced by C-Blu_NI_ and C-Eld_AI_ samples (Figure 6h). Finally, this study showed that despite differences between the overall acceptability of the U-CC samples not being established, U-CC samples induced different intensities of emotions for the judges.

When comparing products with the same acceptability, the emotional profile may aid in differentiation, and this is a trendy topic in sensory science [139]. Emotional profiling of dairy products by different methods was reported in other studies [140,141,142,143]; however, products similar to those in our study were not evaluated in the previous studies. Falkeisen et al. [141] found that higher overall liking scores of plant-based cheeses were associated with positive emotions. The induced emotions for a person by various products varied due to the person’s changing emotional state [144,145]. The emotions “happy” and “surprised” are more often associated with sweet food taste than with others (e.g., salty, sour or bitter) [146,147,148]. It was reported that traditional products, e.g., bread, usually are associated with a “neutral” emotional status [147]. However, new ingredients can induce other emotions. In this study, the emotion “happy” was expressed more intensely for U-CC containing elderberry BIBs. This could be associated with the new experience of testing an unusual formulation of traditional U-CC. It should be noted that the judges were informed about the sustainable ingredients—BIBs—which could be associated with sustainable technological solutions as well as with progress in reducing the problems associated with climate change. Finally, this study showed that despite significant differences between the overall acceptability of the U-CC samples not being found, the induced emotions for judges from various U-CC formulations were varied, and the FaceReader technique could be used as a sensitive method for predicting the market success of new products.

## 4. Conclusions

This study confirmed that the raspberry, blueberry and elderberry BIBs possess desirable antimicrobial and antioxidant activities, and the addition of these by-products to the main U-CC formula increased the TPC content and DPPH- radical scavenging activity of the U-CC. Also, despite the fact that the predominant FAs in U-CC were saturated, the addition of BIBs led to lower saturated FA content in the final product. Additionally, BIBs increased the number of VCs in U-CC, and these changes are associated with the higher intensity of the emotion “happy” induced for consumers from the tested product. However, after 10 days of storage, the highest total enterobacteria number was found in C-Blu_NI_ samples. Finally, this study showed that BIBs are prospective ingredients for U-CC enrichment in a sustainable manner. However, further research is needed to evaluate the possible contamination of BIBs, to avoid non-desirable changes during U-CC production, including microbial contamination and BA formation. Also, future research can be applied to the more detailed analysis of the broader spectrum of BIBs, with the aim of using these valuable by-products in the dairy industry.

## Figures and Tables

**Figure 1 foods-12-02860-f001:**
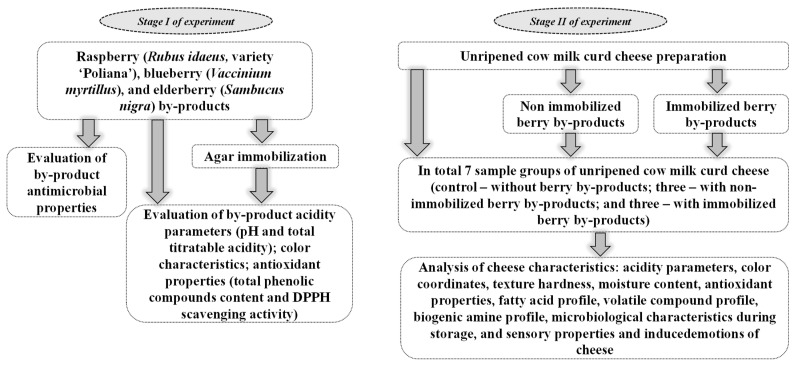
General experimental design of the current study.

**Figure 2 foods-12-02860-f002:**
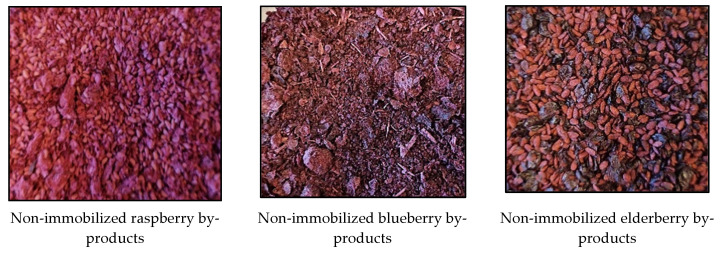
Images of the non-immobilized (_NI_) and agar-immobilized (_AI_) berry industry by-products (BIBs).

**Figure 3 foods-12-02860-f003:**
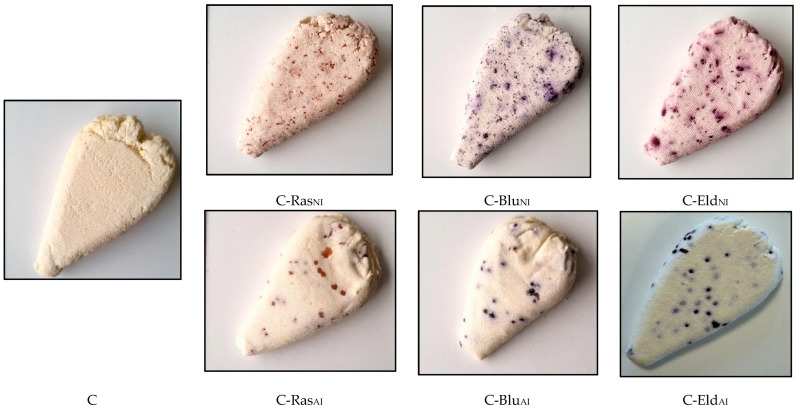
Images of the unripened cow milk curd cheese samples (C—unripened cow milk curd cheese; Ra—raspberry by-products; Blu—blueberry by-products; Eld—elderberry by-products; _NI_—non-immobilized and _AI_—agar-immobilized).

**Figure 4 foods-12-02860-f004:**
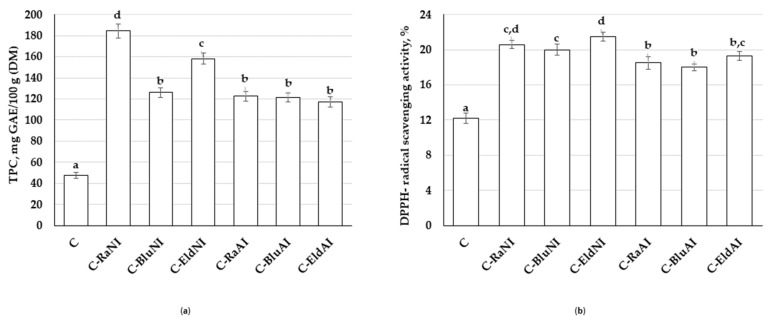
(**a**)—Mean values and standard errors of total content of phenolic compounds (TPC, mg GAE 100/g) and (**b**)—DPPH^−^ radical scavenging activity (%) of unripened cow milk curd cheese samples (C—unripened cow milk curd cheese; Ra—raspberry by-products; Blu—blueberry by-products; Eld—elderberry by-products; _NI_—non-immobilized; _AI_—agar-immobilized; DM—dry matter; TPC—total phenolic compounds; GAE—gallic acid equivalents; DPPH—2,2-diphenyl-1-picrylhydrazyl free radical. Data are expressed as mean values (*n* = 3) ± SE; SE—standard error. a–d—Mean values within columns with different letters are significantly different (*p* ≤ 0.05)).

**Figure 5 foods-12-02860-f005:**
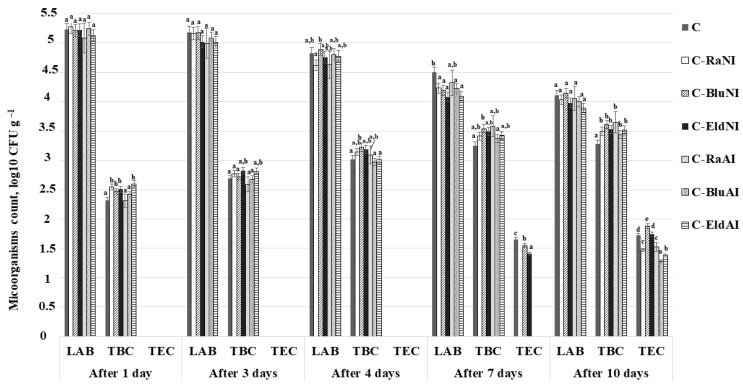
Mean values and standard errors of microbiological parameters (lactic acid bacteria (LAB), total bacteria (TBC) and total enterobacteria (TEC) viable counts) of the unripened cow milk curd cheese (U-CC) after 1, 3, 4, 7 and 10 days of storage (C—unripened cow milk curd cheese; Ra—raspberry by-products; Blu—blueberry by-products; Eld—elderberry by-products; _NI_—non-immobilized; _AI_—agar-immobilized. Data are expressed as mean values (*n* = 3) ± SE; SE—standard error. a–e—mean values with different letters indicate differences among samples (*p* < 0.05).

**Figure 6 foods-12-02860-f006:**
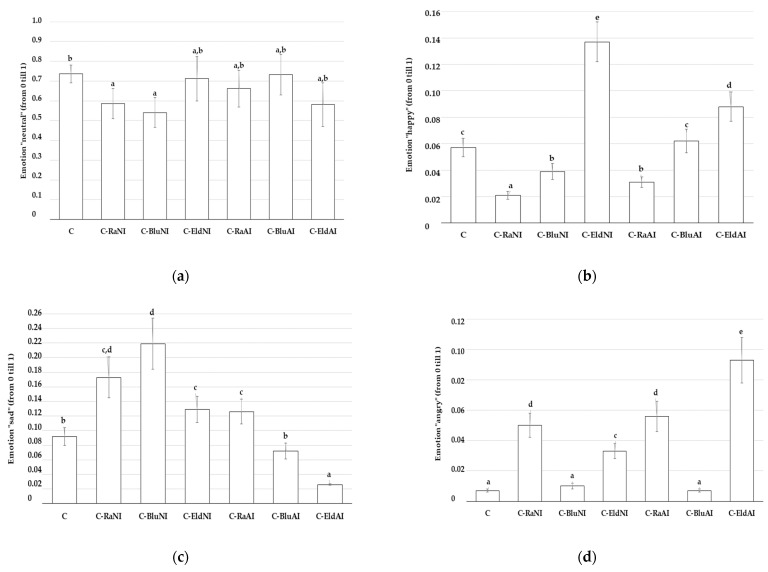
Mean values and standard errors of the intensity of induced emotions for the judges from unripened cow milk curd cheese (U-CC): (**a**–**h**)—intensity of emotions “neutral”, “happy”, “sad”, “angry”, “surprised”, “scared”, “disgusted” and “contempt”, respectively) (C—unripened cow milk curd cheese; Ra—raspberry by-products; Blu—blueberry by-products; Eld—elderberry by-products; _NI_—non-immobilized; _AI_—agar-immobilized; Data are expressed as mean values (*n* = 10) ± SE; SE—standard error. a–e—Mean values within a line with different letters are significantly different (*p* ≤ 0.05)).

**Table 1 foods-12-02860-t001:** Mean values and standard errors of diameter of inhibition zone (DIZ, mm) of berry industry by-products (BIBs) against a set of tested pathogenic and opportunistic bacteria strains.

Berry By-Products	DIZ, mm									
	Pathogenic Opportunistic Bacteria Strains									
	1	2	3	4	5	6	7	8	9	10
Ras	nd	nd	12.0 ± 0.4	15.5 ± 0.3 b	14.4 ± 0.3	12.2 ± 0.2	10.5 ± 0.4 a	13.6 ± 0.4 b	nd	13.4 ± 0.2 b
Blu	nd	9.2 ± 0.2 a	nd	14.2 ± 0.1 a	nd	nd	10.7 ± 0.21 a	12.4 ± 0.3 a	nd	12.3 ± 0.4 a
Eld	nd	13.3 ± 0.4 b	nd	nd	nd	nd	nd	12.9 ± 0.3 a,b	nd	nd

Ras—raspberry by-products; Blu—blueberry by-products; Eld—elderberry by-products; 1—*Salmonella enterica Infantis*; 2—*Staphylococcus aureus*; 3—*E. coli* (hemolytic); 4—*Bacillus pseudomycoides*; 5—*Aeromonas veronii*; 6—*Cronobacter sakazakii*; 7—*Hafnia alvei*; 8—*Enterococcus durans*; 9—*Kluyvera cryocrescens*; 10—*Acinetobacter johnsonii*; nd—not detected. Data are expressed as mean values (*n* = 3) ± SE; SE—standard error. a,b—Mean values within the lines with different letters are significantly different (*p* ≤ 0.05).

**Table 2 foods-12-02860-t002:** Mean values and standard errors of total content of phenolic compounds (mg GAE/100g), DPPH^−^ radical scavenging activity (%), color coordinates (L*, a* and b*, NBS) and acidity parameters of non-immobilized (_NI_) and immobilized (_AI_) berry industry by-products (BIBs).

Berry Industry By-Products	DPPH, %	TPC, mg GAE/100 g (DM)	Color Coordinates, NBS Units			pH	TTA, °N
			L*	a*	b*		
Ras_NI_	75.2 ± 1.47 e	137.4 ± 0.95 e	32.6 ± 3.59 c	29.6 ± 2.31 e	6.75 ± 0.81 d	3.29 ± 0.11 a	0.500 ± 0.010 b
Blu_NI_	68.1 ± 1.33 c	115.1 ± 0.77 b	23.9 ± 2.15 b	10.0 ± 1.11 c	1.88 ± 0.16 c	3.89 ± 0.18 b,c	0.400 ± 0.010 a
Eld_NI_	77.7 ± 1.14 e	143.6 ± 0.89 f	20.2 ± 1.87 a,b	6.07 ± 0.73 b	2.14 ± 0.24 c	4.08 ± 0.21 b,c	0.600 ± 0.010 c
Ras_AI_	64.3 ± 0.95 b	121.7 ± 0.82 c	32.5 ± 3.48 c	20.5 ± 1.84 d	6.83 ± 0.91 d	3.72 ± 0.16 b	0.900 ± 0.020 e
Blu_AI_	61.4 ± 0.78 a	104.3 ± 0.68 a	23.4 ± 2.02 a,b	5.97 ± 0.61 b	0.260 ± 0.020 a	4.22 ± 0.19 c	0.700 ± 0.010 d
Eld_AI_	72.2 ± 1.02 d	130.4 ± 0.85 d	19.9 ± 1.71 a	4.32 ± 0.43 a	1.37 ± 0.09 b	4.67 ± 0.22 d	1.00 ± 0.02 f

Ra—raspberry by-products; Blu—blueberry by-products; Eld—elderberry by-products; _NI_—non-immobilized; _AI_—agar-immobilized; TPC—total phenolic compounds; GAE—gallic acid equivalents; DPPH—2,2-diphenyl-1-picrylhydrazyl free radical; TTA—total titratable acidity; L* lightness; a* redness or -a* greenness; b* yellowness or -b* blueness; NBS—National Bureau of Standards units. Data are expressed as mean values (*n* = 3) ± SE; SE—standard error. a–f—Mean values within a line with different letters are significantly different (*p* ≤ 0.05).

**Table 3 foods-12-02860-t003:** Mean values and standard errors of pH and acidity (TTA) parameters, color (L*, a* and b*) characteristics and texture parameters of unripened cow milk curd cheese (U-CC).

Cheese Samples	Acidity Parameters		Color Coordinates, NBS Units			Texture, mJ	Moisture, %
	pH	TTA, °T	L*	a*	b*		
C	5.80 ± 0.02 d	1.90 ± 0.12 a	99.5 ± 3.48 e	−4.68 ± 0.47 a	25.0 ± 1.64 d	0.100 ± 0.010 a	68.4 ± 2.4 a
C-Ra_NI_	5.27 ± 0.03 a	2.20 ± 0.14 b	89.8 ± 2.47 d	1.07 ± 0.22 d	18.6 ± 1.18 c	0.300 ± 0.010 c	68.2 ± 2.1 a
C-Blu_NI_	5.35 ± 0.02 b	2.30 ± 0.17 b	80.4 ± 3.21 c	1.69 ± 0.39 e	8.26 ± 0.51 a	0.200 ± 0.010 b	68.1 ± 1.9 a
C-Eld_NI_	5.58 ± 0.02 c	2.50 ± 0.22 b	70.2 ± 3.82 a	7.35 ± 0.88 f	7.24 ± 0.49 a	0.300 ± 0.010 c	67.8 ± 1.6 a
C-Ra_AI_	5.31 ± 0.01 a	2.40 ± 0.21 b	90.0 ± 4.36 d	−2.71 ± 0.44 b	20.8 ± 1.49 c	0.500 ± 0.010 e	67.5 ± 1.2 a
C-Blu_AI_	5.39 ± 0.02 b	2.30 ± 0.19 b	78.7 ± 2.90 b	−1.94 ± 0.32 c	15.0 ± 1.15 b	0.400 ± 0.010 d	67.9 ± 1.7 a
C-Eld_AI_	5.61 ± 0.07 c	2.40 ± 0.23 b	70.5 ± 2.58 a	2.12 ± 0.29 e	7.9 ± 0.40 a	0.500 ± 0.020 e	67.7 ± 1.4 a

C—unripened cow milk curd cheese; Ra—raspberry by-products; Blu—blueberry by-products; Eld—elderberry by-products; _NI_—non-immobilized; _AI_—agar-immobilized; TTA—total titratable acidity; L* lightness; a* redness or -a* greenness; b* yellowness or -b* blueness; NBS—National Bureau of Standards units. Data are expressed as mean values (*n* = 3) ± SE; SE—standard error. a–f—Mean values within a line with different letters are significantly different (*p* ≤ 0.05).

**Table 4 foods-12-02860-t004:** Mean values and standard errors of fatty acid (FA) profile (% from the total fatty acid content) of unripened cow milk curd cheese (U-CC).

Fatty Acid	C	C-Ra_NI_	C-Blu_NI_	C-Eld_NI_	C-Ra_AI_	C-Blu_AI_	C-Eld_AI_
	Fatty acid content, % from the total fat content						
Butyric acid	3.42 ± 0.34 d	2.84 ± 0.13 b	2.74 ± 0.15 b	3.93 ± 0.22 d	3.87 ± 0.21 d	3.29 ± 0.22 c	2.05 ± 0.14 a
Caproic acid	1.97 ± 0.11 c	1.49 ± 0.07 b	1.63 ± 0.08 b	2.30 ± 0.14 d	2.25 ± 0.15 d	1.93 ± 0.09 c	1.21 ±0.06 a
Caprylic acid	0.720 ± 0.027 d	0.290 ± 0.016 a	0.660 ± 0.023 c	0.990 ± 0.038 f	1.13 ± 0.05 g	0.900 ± 0.029 e	0.470 ± 0.018 b
Capric acid	2.65 ± 0.17 b,c	2.12 ± 0.11 b	2.35 ± 0.12 b	3.22 ± 0.27 d	3.30 ± 0.23 d	2.63 ± 0.16 b,c	1.68 ± 0.18 a
Lauric acid	3.14 ± 0.18 c	2.50 ± 0.12 b	2.59 ± 0.13 b	3.51 ± 0.22 c	3.41 ± 0.19 c	2.78 ± 0.12 b	1.77 ± 0.11 a
Myristic acid	11.4 ± 0.44 c	10.3 ± 0.37 b	9.74 ± 0.31 b	12.2 ± 0.44 c	11.4 ± 0.41 c	9.94 ± 0.34 b	6.70 ± 0.21 a
Myristoleic acid	0.880 ± 0.032 e	0.620 ± 0.023 b	0.710 ± 0.030 c	0.970 ± 0.039 f	0.996 ± 0.040 f	0.820 ± 0.028 d	0.520 ± 0.021 a
Pentadecylic acid	1.07 ± 0.06 e	0.710 ± 0.022 c	0.860 ± 0.027 d	1.14 ± 0.08 e,f	0.100 ± 0.004 a	0.990 ± 0.032 e	0.610 ± 0.021 b
Palmitic acid	33.8 ± 0.75 d	30.8 ± 0.82 b,c	29.1 ± 0.60 b	33.7 ± 0.72 d	32.2 ± 0.69 c	29.6 ± 0.61 b	21.5 ± 0.53 a
Margaric acid	0.450 ± 0.011 e	0.270 ± 0.004 b	0.360 ± 0.007 c	0.430 ± 0.010 d	0.460 ± 0.012 e	0.380 ± 0.006 c	0.220 ± 0.004 a
Palmitoleic acid	1.61 ± 0.08 c	1.32 ± 0.07 b	1.33 ± 0.06 b	1.69 ± 0.09 c,d	1.72 ± 0.10 c,d	1.48 ± 0.07 c	1.06 ± 0.05 a
Stearic acid	11.5 ± 0.45 c	10.4 ± 0.40 b	10.2 ± 0.38 b	11.4 ± 0.43 b,c	10.6 ± 0.39 b	10.4 ± 0.37 b	7.31 ± 0.11 a
Oleic acid	22.1 ± 0.70 a	21.6 ± 0.77 a	21.1 ± 0.73 a	21.8 ± 0.75 a	24.0 ± 0.78 b	25.5 ± 0.81 b	37.6 ± 0.87 c
Linoleic acid	4.23 ± 0.17 c	10.7 ± 0.27 f	5.74 ± 0.19 d	2.12 ± 0.11 a	2.98 ± 0.13 b	6.49 ± 0.22 e	11.2 ± 0.29 f
α-Linolenic acid	1.09 ± 0.03 c	4.78 ± 0.11 e	8.04 ± 0.25 f	0.630 ± 0.015 a	0.880 ± 0.019 b	2.40 ± 0.09 d	5.00 ± 0.15 e
Arachidic acid	nd	nd	0.190 ± 0.007 b	nd	nd	0.090 ± 0.004 a	0.200 ± 0.011 b
Gondoic acid	nd	nd	2.74 ± 0.17 e	0.030 ± 0.002 a	0.070 ± 0.004 b	0.510 ± 0.021 c	0.800 ± 0.024 d
	Fatty acid profile, % from the total fat content						
Omega-3	1.09 ± 0.06 c	4.78 ± 0.13 e	8.04 ± 0.26 f	0.630 ± 0.011 a	0.880 ± 0.018 b	2.40 ± 0.11 d	5.00 ± 0.14 e
Omega-6	4.23 ± 0.11 c	10.7 ± 0.16 f	5.74 ± 0.13 d	2.12 ± 0.12 a	2.98 ± 0.14 b	6.49 ± 0.32 e	11.2 ± 0.52 f
Omega-9	22.1 ± 0.61 a	21.6 ± 0.64 a	23.8 ± 0.75 b	21.8 ± 0.65 a	23.5 ± 0.70 b	26.0 ± 0.75 c	38.4 ± 0.85 d
SFA	70.1 ±0.88 d	61.0 ± 0.66 b	60.4 ± 0.67 b	72.8 ± 0.77 e	69.9 ± 0.91 d	62.8 ± 0.82 c	43.7 ± 0.69 a
MUFA	24.6 ± 0.56 a	23.5 ± 0.52 a	25.8 ± 0.59 b	24.5 ± 0.58 a	26.3 ± 0.61 b	28.3 ± 0.64 c	40.0 ± 0.73 d
PUFA	5.31 ± 0.15 c	15.5 ± 0.29 f	13.8 ± 0.22 e	2.75 ± 0.11 a	3.86 ± 0.14 b	8.88 ± 0.19 d	16.2 ± 0.32 g
**Multivariate Test Results**							
**Fatty acid**		**Significance (*p*) of the influence of analyzed factors and their interaction on FA content in U-CC samples**					
		**Type of BIB**		**Immobilization**		**Interaction: type of BIB * immobilization**	
Butyric acid		**0.018**		0.335		**<0.001**	
Caproic acid		0.365		0.457		0.465	
Caprylic acid		0.155		0.078		0.294	
Capric acid		0.909		0.178		**0.035**	
Lauric acid		0.625		0.205		0.158	
Myristic acid		**0.028**		0.378		0.575	
Myristoleic acid		0.231		0.145		0.080	
Pentadecylic acid		0.356		0.103		**0.025**	
Palmitic acid		**<0.001**		**0.007**		**<0.001**	
Margaric acid		0.225		0.320		0.827	
Palmitoleic acid		**0.003**		0.459		**<0.001**	
Stearic acid		0.203		**0.010**		0.156	
Oleic acid		0.314		0.671		0.495	
Linoleic acid		0.675		0.655		**0.044**	
α-Linolenic acid		0.058		0.253		**0.029**	
Arachidic acid		**<0.001**		**0.011**		**<0.001**	
Gondoic acid		0.084		0.041		0.126	
Omega-3		0.193		0.399		0.318	
Omega-6		0.851		0.558		**0.028**	
Omega-9		**<0.001**		**<0.001**		**<0.001**	
SFA		**0.035**		0.206		**0.006**	
MUFA		0.105		0.675		**0.019**	
PUFA		**<0.001**		**<0.001**		**<0.001**	

C—unripened cow milk curd cheese; Ra—raspberry by-products; Blu—blueberry by-products; Eld—elderberry by-products; _NI_—non-immobilized; _AI_—agar-immobilized; nd—not detected; SFA—saturated fatty acid; MUFA—monounsaturated fatty acids; PUFA—polyunsaturated fatty acid; FA—fatty acid; U-CC—unripened cow milk curd cheese; BIB—berry industry by-product; *—interaction of analyzed factors. Data are expressed as mean values (*n* = 3) ± SE; SE—standard error. a–g—Mean values within a line with different letters are significantly different (*p* ≤ 0.05). Factors and their interaction are significant when *p* ≤ 0.05. Numbers marked in Bold are significant.

**Table 5 foods-12-02860-t005:** Mean values and standard errors of volatile compounds (VC) (% from the total volatile compounds content) of unripened cow milk curd cheese (U-CC).

RT, min	Volatile Compound	Cheese Samples						
		C	C-Ra_NI_	C-Blu_NI_	C-Eld_NI_	C-Ra_AI_	C-Blu_AI_	C-Eld_AI_
	Aldehydes							
6.73	Hexanal	nd	7.98 ± 0.67 c	1.47 ± 0.13 b	nd	0.310 ± 0.031 a	nd	nd
11.73	(E)-hept-2-enal	nd	0.480 ± 0.011	nd	nd	nd	nd	nd
11.86	Benzaldehyde	nd	0.710 ± 0.019 a	nd	0.710 ± 0.017 a	nd	nd	nd
14.59	Benzeneacetaldehyde	nd	nd	nd	0.440 ± 0.009	nd	nd	nd
16.53	Nonanal	0.480 ± 0.031 a	1.31 ± 0.16 c	0.490 ± 0.090 a	0.410 ± 0.080 a	0.480 ± 0.100 a	0.770 ± 0.070 b	0.710 ± 0.081 b
	Ketones							
9.56	2-Heptanone	2.84 ± 0.26 d	1.18 ± 0.11 a,b	2.55 ± 0.19 d	2.77 ± 0.22 c	1.34 ± 0.14 b	2.02 ± 0.18 c	1.02 ± 0.12 a
14.43	3-Octen-2-one	nd	0.090 ± 0.012	nd	nd	nd	nd	nd
15.46	3,5-Octadien-2-one	nd	1.41 ± 0.21	nd	nd	nd	nd	nd
16.14	2-Nonanone	3.71 ± 0.35 b	3.54 ± 0.31 b	3.42 ± 0.27 b	3.38 ± 0.26 b	2.66 ± 0.15 a	2.65 ± 0.18 a	3.04 ± 0.22 a,b
22.12	2-Undecanone	1.39 ± 0.11 d	0.940 ± 0.017 b	1.11 ± 0.08 c	1.42 ± 0.12 d	0.890 ± 0.015 a	1.11 ± 0.07 c	1.00 ± 0.05 b,c
27.41	2-Tridecanone	0.520 ± 0.02 b	0.620 ± 0.031 c	0.920 ± 0.041 d	0.590 ± 0.033 c	0.570 ± 0.029 b,c	0.370 ± 0.017 a	0.590 ± 0.034 c
	Terpenoids							
11.06	α-Pinene	0.440 ± 0.020 c	0.210 ± 0.012 a	0.370 ± 0.016 b	0.410 ± 0.023 c	0.373 ± 0.018 b	0.520 ± 0.024 d	0.370 ± 0.017 b
12.35	Sabinene	0.410 ± 0.018 d	0.230 ± 0.011 a	0.360 ± 0.012 b	0.390 ± 0.013 c	0.490 ± 0.017 e	0.510 ± 0.022 e	0.480 ± 0.019 e
12.47	β-Pinene	4.33 ± 0.31 a,b	3.83 ± 0.25 a	4.94 ± 0.35 b	4.99 ± 0.37 b	6.27 ± 0.42 d	5.44 ± 0.39 b,c	5.05 ± 0.37 c
12.93	β-myrcene	1.09 ± 0.07 a	nd	1.93 ± 0.11 c	1.66 ± 0.10 b	1.72 ± 0.12 b,c	2.01 ± 0.15 c	1.94 ± 0.13 c
14.03	p-Cymene	1.22 ± 0.09 b,c	0.920 ± 0.030 a	1.33 ± 0.07 c	1.08 ± 0.05 b	1.22 ± 0.10 b,c	1.03 ± 0.06 b	1.57 ± 0.14 c
14.14	D-Limonene	57.8 ± 2.9 c	36.2 ± 1.2 a	50.8 ± 1.8 b	55.5 ± 2.8 c	58.8 ± 2.9 c	57.9 ± 2.7 c	57.9 ± 2.6 c
15.13	γ-Terpinene	7.22 ± 0.27 a	8.05 ± 0.30 b	7.18 ± 0.23 a	6.77 ± 0.18 a	8.97 ± 0.25 c	8.00 ± 0.27 b,c	7.57 ± 0.29 b
16.38	Linalol	0.270 ± 0.012 b	0.340 ± 0.021 c	0.690 ± 0.031 e	0.440 ± 0.022 d	0.380 ± 0.019 c	0.280 ± 0.014 b	0.240 ± 0.017 a
17.87	trans-Verbenol	nd	nd	nd	nd	0.120 ± 0.012	nd	nd
18.85	Terpinen-4-ol	0.390 ± 0.013 a	nd	nd	0.591 ± 0.021 d	0.520 ± 0.019 c	0.470 ± 0.020 b	0.330 ± 0.018 a
19.25	α-Terpineol	0.490 ± 0.022 c	0.440 ± 0.018 b	0.720 ± 0.028 d	0.390 ± 0.017 a	0.440 ± 0.021 b	2.04 ± 0.11 f	1.80 ± 0.05 e
19.83	Verbenone	nd	nd	nd	nd	0.310 ± 0.031	nd	nd
21.50	Citral	0.520 ± 0.022 c,d	0.620 ± 0.025 e	0.580 ± 0.021 d	0.490 ± 0.017 c	0.470 ± 0.015 c	0.290 ± 0.013 b	0.120 ± 0.009 a
22.01	Isobornyl acetate	nd	nd	nd	nd	0.550 ± 0.060	nd	nd
24.54	Geranyl acetate	nd	nd	0.080 ± 0.006 a	nd	0.120 ± 0.011 b	0.090 ± 0.009 a	nd
25.74	Caryophyllene	0.220 ± 0.021 c,d	0.170 ± 0.013 c	0.140 ± 0.010 b	0.190 ± 0.012 c	0.290 ± 0.022 e	0.120 ± 0.010 b	0.100 ± 0.007 a
26.03	cis-α-Bergamotene	0.330 ± 0.022 c	0.270 ± 0.014 b	0.310 ± 0.024 c	0.240 ± 0.013 a	nd	0.520 ± 0.021 d	0.510 ± 0.025 d
26.09	Calarene	nd	nd	nd	nd	2.01 ± 0.21	nd	nd
27.84	β-Bisabolene	0.230 ± 0.012 b	0.250 ± 0.017 b,c	0.670 ± 0.032 e	0.210 ± 0.014 b	0.350 ± 0.021 d	0.730 ± 0.035 f	0.170 ± 0.009 a
	Organic acids							
12.73	Hexanoic acid	3.99 ± 0.19 d	13.69 ± 1.36 g	5.92 ± 0.47 f	4.91 ± 0.36 e	1.77 ± 0.14 a	3.02 ± 0.17 c	2.07 ± 0.11 b
15.61	Heptanoic acid	nd	0.270 ± 0.014	nd	nd	nd	nd	nd
18.72	Octanoic acid	6.87 ± 0.12 d	9.33 ± 0.22 f	7.98 ± 0.17 e	6.98 ± 0.77 d	3.44 ± 0.11 a	4.90 ± 0.14 c	3.89 ± 0.13 b
24.10	Decanoic acid	2.39 ± 0.11 c	4.11 ± 0.18 e	3.33 ± 0.17 d	3.03 ± 0.24 d	1.98 ± 0.15 b	3.08 ± 0.15 d	1.65 ± 0.11 a
	Aliphatic hydrocarbons							
13.20	Decane	nd	nd	nd	nd	1.33 ± 0.05 a	nd	3.14 ± 0.11 b
18.36	5-(2-Methylpropyl)nonane	nd	nd	nd	nd	nd	nd	0.060 ± 0.021
19.43	Dodecane	1.59 ± 0.13 c,d	1.54 ± 0.11 c	1.39 ± 0.10 c	1.07 ± 0.06 b	0.880 ± 0.027 a	0.890 ± 0.029 a	3.08 ± 0.27 c
19.85	6-Methyldodecane	nd	nd	nd	nd	nd	nd	0.160 ± 0.011
21.74	4,6-dimethyldodecane	0.380 ± 0.017 d	0.250 ± 0.013 a	0.290 ± 0.011 b	0.320 ± 0.022 b,c	0.290 ± 0.012 b	0.470 ± 0.032 e	0.340 ± 0.021 b,c
24.95	Tetradecane	0.540 ± 0.031 c	0.480 ± 0.025 b	0.590 ± 0.034 c,d	0.640 ± 0.037 d	0.470 ± 0.024 a	0.550 ± 0.033 c	0.370 ± 0.018 a
	Other compounds							
13.75	3,4-Dimethylbenzyl alcohol	nd	nd	nd	nd	nd	nd	0.180 ± 0.016
21.07	1,3-bis(1,1-dimethylethyl)benzene	0.313 ± 0.013 c	0.350 ± 0.015 d	0.430 ± 0.027 e	0.240 ± 0.017 b	0.190 ± 0.014 a	0.220 ± 0.016 a,b	0.550 ± 0.029 f
24.84	Decanoic acid, ethyl ester	nd	0.190 ± 0.014 b	nd	0.090 ± 0.008 a	nd	nd	nd

C—unripened cow milk curd cheese; Ra—raspberry by-products; Blu—blueberry by-products; Eld—elderberry by-products; _NI_—non-immobilized; _AI_—agar-immobilized; RT—retention time; nd—not detected. Data are expressed as mean values (*n* = 3) ± SE; SE—standard error. a–g—Mean values within a line with different letters are significantly different (*p* ≤ 0.05).

**Table 6 foods-12-02860-t006:** Significance (*p*) of the influence of analyzed factors and their interaction on volatile compound (VC) content of unripened cow milk curd cheese (U-CC).

Volatile Compound	Significance (*p*) of the Influence of Analyzed Factors and Their Interaction on VC Content in U-CC Samples		
	Type of BIB	Immobilization	Interaction: Type of BIB * Immobilization
Hexanal	**<0.001**	**<0.001**	**<0.001**
2-Heptanone	0.234	**0.006**	0.144
α-Pinene	0.082	**0.033**	**0.021**
(E)-hept-2-enal	**0.012**	**0.025**	**0.011**
Benzaldehyde	**0.025**	**0.003**	0.073
Sabinene	0.941	0.277	0.893
β-Pinene	**0.008**	**0.004**	**0.048**
Hexanoic acid	**<0.001**	**<0.001**	**<0.001**
β-Myrcene	0.241	0.445	0.258
Decane	**<0.001**	**<0.001**	**<0.001**
3,4-Dimethylbenzyl alcohol	**0.027**	**0.026**	**0.012**
p-Cymene	0.495	0.500	0.338
D-Limonene	**<0.001**	**<0.001**	**<0.001**
3-Octen-2-one	**0.015**	**0.029**	**0.013**
Benzeneacetaldehyde	0.064	0.076	0.053
γ-Terpinene	0.219	0.968	0.337
3,5-Octadien-2-one	**0.016**	**0.030**	**0.014**
Heptanoic acid	**0.012**	**0.025**	**0.011**
2-Nonanone	0.326	**<0.001**	0.194
Linalol	0.582	0.202	0.452
Nonanal	**0.012**	0.833	0.575
trans-Verbenol	**0.028**	**0.027**	**0.012**
5-(2-Methylpropyl)nonane	0.242	0.148	0.132
Octanoic acid	0.156	**<0.001**	0.091
Terpinen-4-ol	0.065	0.395	0.067
α-Terpineol	0.693	0.186	0.535
Dodecane	0.577	**0.005**	0.557
Verbenone	**<0.001**	**<0.001**	**<0.001**
6-Methyldodecane	**0.027**	**0.026**	**0.012**
1,3-bis(1,1-dimethylethyl)benzene	0.868	0.875	0.206
Citral	0.891	0.062	0.979
4,6-dimethyldodecane	0.831	0.525	0.850
Isobornyl acetate	**<0.001**	**<0.001**	**<0.001**
2-Undecanone	0.484	0.262	0.407
Decanoic acid	**0.045**	**0.030**	**0.017**
Geranyl acetate	**0.045**	0.078	0.092
Decanoic acid, ethyl ester	**0.023**	**0.005**	**0.048**
Tetradecane	0.977	0.574	0.824
Caryophyllene	0.063	0.903	0.383
cis-α-Bergamotene	0.262	**0.044**	0.052
Calarene	0.398	0.228	0.239
2-Tridecanone	0.607	0.368	0.529
β-Bisabolene	0.306	0.397	0.599

VC—volatile compound; U-CC—unripened cow milk curd cheese; BIB—berry industry by-product; *—interaction of analyzed factors. Factors and their interaction are significant when *p* ≤ 0.05. Numbers marked in Bold are significant.

**Table 7 foods-12-02860-t007:** Mean values and standard errors of biogenic amine (BA) content (mg/kg) of unripened cow milk curd cheese (U-CC).

Biogenic Amines Content, mg/kg								
Cheese Samples	TRY	PHE	PUT	CAD	HIS	TYR	SPER	SPRMD
C	nd	nd	nd	nd	nd	nd	nd	nd
C-Ra_NI_	nd	nd	nd	nd	nd	nd	nd	2.40 ± 0.28 b
C-Blu_NI_	nd	nd	nd	nd	nd	nd	nd	11.6 ± 1.38 c
C-Eld_NI_	nd	nd	7.96 ± 1.03 a	nd	nd	nd	nd	1.40 ± 0.14 a
C-Ra_AI_	nd	nd	nd	nd	nd	nd	nd	nd
C-Blu_AI_	nd	nd	nd	nd	nd	nd	nd	nd
C-Eld_AI_	0.870 ± 0.110	nd	22.6 ± 2.48 b	19.8 ± 2.77	8.91 ± 1.15	11.2 ± 1.23	21.0 ± 1.89	19.7 ± 2.37 d

C—unripened cow milk curd cheese; Ra—raspberry by-products; Blu—blueberry by-products; Eld—elderberry by-products; _NI_—non-immobilized; _AI_—agar-immobilized; TRY—tryptamine; PHE—phenylethylamine; PUT—putrescine; CAD—cadaverine; HIS—histamine; TYR—tyramine; SPER—spermine; SPRMD—spermidine; nd—not detected. Data are expressed as mean values (*n* = 3) ± SE; SE—standard error. a–d—Mean values within a line with different letters are significantly different (*p* ≤ 0.05).

## Data Availability

The data presented in this study are available on request from the corresponding author.

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
