# Peer review of "Characteristics of Unripened Cow Milk Curd Cheese Enriched with Raspberry (Rubus idaeus), Blueberry (Vaccinium myrtillus) and Elderberry (Sambucus nigra) Industry By-Products"

_foods, 2023, doi:10.3390/foods12152860_

Round 1

Reviewer 1 Report

The aim of the submitted work was to produce and analyze unripened curd cheeses fortified with different fruit industry by-products. Submitted manuscript deals with the development of functional dairy products in terms of improvement of their antioxidant and nutritional value (e.g. increased dietary  fiber and unsaturated fatty acids concentrations) and at the same time with utilization of by-products of the fruit industry. In that light, the topic is interesting and of high utilization potential.

The manuscript is well written, the methods used appropriate. However, I have some comments and suggest minor changes, which are listed below:

11. Abstract & Abbreviations  - there is no explanation for some abbreviations, e.g. C-EldAI

22.     Keywords – I suggest to remove “nutrition” as nutrition was not the subject of the study

33.     Introduction – rewrite the sentence in the lines 88-91 (“which are used not efficiently used so far”?)

44.     Introduction – line 104 – “flavanols” are mentioned two times

55.     Introduction – line 108 – “Elderberries has”?

66.     Materials and methods – line 180 – provide information regarding amount of lemon juice used for milk coagulation and/or the final pH of coagulation

77.     Materials and methods – lines 191-192 – clarify how the curd was mixed, cut and drained (Was it any interval between mixing and cutting?). It is also not clear how the cheese mass was pressed?

88.     Methods – the main reason of the microbiological spoilage of cheese are yeast and molds – why yeast and molds were not enumerated during the study?

99.    Results and discussion – lines 489-491 – the given values for L* (in %?) do not correspond with the values of lightness given in Table 3 – please clarify

110.   Results and discussion – lines 555-557 – the authors wrote that analyzed factors, such as type of  BIB and immobilization were not statistically significant on TPC content and DPPH radical scavenging activity but on the other hand there are different letters in Fig 4 suggesting significant differences between selected results – please clarify

111.   Results and discussion – where are the results or the overall sensory acceptability of produced cheeses?

112.   Results and discussion – I wonder what was the reason of microbial contamination of cheeses especially during storage, e.g. storage conditions/packagings (were the cheeses wrapped)?

Language should be improved throughout the manuscript.

Author Response

Dear Reviewer, below You will find Authors responses.

Reviewer 1:

The aim of the submitted work was to produce and analyze unripened curd cheeses fortified with different fruit industry by-products. Submitted manuscript deals with the development of functional dairy products in terms of improvement of their antioxidant and nutritional value (e.g. increased dietary  fiber and unsaturated fatty acids concentrations) and at the same time with utilization of by-products of the fruit industry. In that light, the topic is interesting and of high utilization potential.

Authors response: Authors are thankful for the Reviewer evaluation and comment.

The manuscript is well written, the methods used appropriate. However, I have some comments and suggest minor changes, which are listed below:

Reviewer 1:

  1. Abstract & Abbreviations - there is no explanation for some abbreviations, e.g. C-EldAI

Authors response: Authors are thankful for Reviewer comment, however, the authors would like to kindly explain, that in the Abstract section all abbreviations are explained:

The aim of this study was to apply raspberry (Ras), blueberry (Blu), and elderberry (Eld) industry by-products (BIB) for unripened cow milk curd cheese (U-CC) enrichment. Firstly, antimicrobial properties of the BIBs were tested and the effects of the immobilization in agar technology on BIB properties were evaluated. Further, non-immobilized (NI) and agar-immobilized (AI) BIBs were applied for U-CC enrichment, and their influence on U-CC parameters were analyzed.

Reviewer 1:

  1. Keywords – I suggest to remove “nutrition” as nutrition was not the subject of the study

Authors response: Authors are thankful for comment, and ‘nutrition‘ was removed from Keywords list.

Keywords: unripened cow milk curd cheese; berry industry by-product; antimicrobial properties; antioxidant characteristics; biogenic amine; volatile compounds.

Reviewer 1:

  1. Introduction – rewrite the sentence in the lines 88-91 (“which are used not efficiently used so far”?)

Authors response: Authors are thankful for Reviewer comment, the sentence in the mentioned lines was corrected.

Industrial processes (juice, wine, etc. production) greatly contribute to BIB production, which, till now, are used not enough efficient.

Reviewer 1:

  1. Introduction – line 104 – “flavanols” are mentioned two times

Authors response: Authors are thankful for valuable comment, corrected.

Additionally, the blueberries are an abundant source of sugars (glucose and fructose), vitamins, folic acid, minerals, organic acids, flavanols, and anthocyanins [11–15].

Reviewer 1:

  1. Introduction – line 108 – “Elderberries has”?

Authors response: Authors are thankful for valuable Reviewer comment, the sentence has been corrected.

Elderberries beneficial effect on human health is widely described [16,18,19].

Reviewer 1:

  1. Materials and methods – line 180 – provide information regarding amount of lemon juice used for milk coagulation and/or the final pH of coagulation

Authors response: Authors are thankful for Reviewer comment, the information has been included.

For coagulation, pure organic lemon juice was used (30 mL per litre of milk), in addition to 0.2 g/L of milk CaCl2 were added.

Reviewer 1:

  1. Materials and methods – lines 191-192 – clarify how the curd was mixed, cut and drained (Was it any interval between mixing and cutting?). It is also not clear how the cheese mass was pressed?

Authors response: Authors are thankful for Reviewer comment, the additional information was included, and also, authors would like to explain, that there was no intervals between mixing and cutting.

2.2.2. Preparation of unripened cow milk curd cheese (U-CC)

The U-CC was prepared using 18 L of raw cow milk per treatment. Raw cow milk intended for U-CC was pasteurized at (72-73 °C) for (15-20 s), followed by cooling to (30 ± 2 °C). Then, 1% (w/v) of non-immobilized and 4% (w/v) of immobilized BIB were added. For coagulation, pure organic lemon juice was used (30mL per litre of milk), in addition to 0.2 g/L of milk CaCl2 were added. After milk coagulation, curd was mixed, using a wooden spoon, and gently cut into 200 g cubes and drained. Furthermore, mass was placed in nylon containers and pressed (0.4 kg weight) for 12 h at 4 °C. U-CC samples without BIB were analyzed as a control.

Reviewer 1:

  1. Methods – the main reason of the microbiological spoilage of cheese are yeast and molds – why yeast and molds were not enumerated during the study?

Authors response: Authors are thankful for valuable Reviewer comment. We would like to explain, that yeast and mold number was evaluated, however, during the storage period (10 days) no visible colonies were not detected, for this reason, these results were not included.

Reviewer 1:

  1. Results and discussion – lines 489-491 – the given values for L* (in %?) do not correspond with the values of lightness given in Table 3 – please clarify

Authors response: Authors are thankful for Reviewer comment, the lines has been corrected.

Other U-CC sample L* values were lower (on average, by 29.2% for C-EldAI, and, on average, by 9.67% for C-RaNI and C-RaAI), in comparison with control samples.

Reviewer 1:

  1. Results and discussion – lines 555-557 – the authors wrote that analyzed factors, such as type of BIB and immobilization were not statistically significant on TPC content and DPPH radical scavenging activity but on the other hand there are different letters in Fig 4 suggesting significant differences between selected results – please clarify

Authors response: Authors are thankful for comment. We would like to explain, that for these calculations different statistic methods were used and significant differences between the values does not means that factors (ot their interaction) are significant of these parameters. When food technologies are developed, many factors can be involved during the process, however, not all of it are identified (this is not possible), for this reason, significant different values between the parameters, does not mean, that analysed factors are significant on these parameter. For this reason, it is very important to calculate influence on analyse factors on concrete parameters, which are of most interest.

Reviewer 1:

  1. Results and discussion – where are the results or the overall sensory acceptability of produced cheeses?

Authors response: Authors are thankful for Reviewer comment, the authors would like to explain, that the results of overall acceptability are described in section 3.2.6.:

Significant differences between U-CC overall acceptability were not obtained and, on average, overall acceptability of the U-CC was 8.34 points.

Reviewer 1:

  1. Results and discussion – I wonder what was the reason of microbial contamination of cheeses especially during storage, e.g. storage conditions/packagings (were the cheeses wrapped)?

Authors response: Authors are thankful for the Reviewer comment, we agree that further research is needed, therefore, and additional suggestions for further research were included:

         Finally, this study showed that BIBs are prospective ingredients for U-CC enrichment in sustainable manner. However, further research is needed to evaluate possible contamination of the BIBs, to avoid non-desirable changes during the U-CC production, including microbial contamination and BA formation. Also, future research can be applied to the more detailed analysis of the broader spectrum of BIBs, with the aim to use these valuable by-products in dairy industry.

Reviewer 2 Report

The authors provided interesting research concerning the application of raspberry, blueberry, and elderberry industry by-products for the unripened cow milk curd cheese (U-CC) enrichment. The manuscript is well organized and the matrices as well as the final products were deeply characterized. 

Author Response

Dear Reviewer,

Authors are thankful for the evaluation and comment.

Reviewer 3 Report

Authors did significant work in the manuscript entitled “Characteristics of unripened cow milk curd cheese enriched with raspberry (Rubus idaeus), blueberry (Vaccinium myrtillus) and elderberry (Sambucus nigra) industry by-products”. However, there are some minor issues within the manuscript that needs to be resolved.

Abstract

Please include the numerical values in this section so that one can easily understand the output of your work.

Materials and methods section

2.3.1. Evaluation of the antimicrobial properties in berry industry by-products (BIB)

Please cite the standard method followed. Authors didn’t mention the quantity of liquid sample used during the concerned assay. Please add.

2.3.3. Determination of total phenolic compounds (TPC) and 2, 2-diphenyl-1-picrylhydrazyl (DPPH- )-radical scavenging activity

Information regarding the concentration of sodium carbonate used during the assay is missing in the concerned section.

Further, information related to DPPH reagent has not been mentioned. What about the use of extraction phase/solvent during the assay which was used for dissolving DPPH? What was the concentration of DPPH used? Absorbance details are also missing.

Authors represent the observations from their results but didn’t compare with the latest scientific reports. Further, reason behind the specific observations has not been mentioned. Please clarify.

Author Response

Dear Reviewer, below You will find Authors responses.

Reviewer 3:

Authors did significant work in the manuscript entitled “Characteristics of unripened cow milk curd cheese enriched with raspberry (Rubus idaeus), blueberry (Vaccinium myrtillus) and elderberry (Sambucus nigra) industry by-products”. However, there are some minor issues within the manuscript that needs to be resolved.

Authors response: Authors are thankful for valuable comments.

Reviewer 3:

Abstract

Please include the numerical values in this section so that one can easily understand the output of your work.

Authors response: Authors are thankful for comment, numerical values were included.

The aim of this study was to apply raspberry (Ras), blueberry (Blu), and elderberry (Eld) industry by-products (BIB) for unripened cow milk curd cheese (U-CC) enrichment. Firstly, antimicrobial properties of the BIBs were tested and the effects of the immobilization in agar technology on BIB properties were evaluated. Further, non-immobilized (NI) and agar-immobilized (AI) BIBs were applied for U-CC enrichment, and their influence on U-CC parameters were analyzed. It was established that the tested BIBs possess desirable antimicrobial (raspberry BIB inhibited 7 out of 10 tested pathogens) and antioxidant activities (the highest total phenolic compounds (TPC) content was displayed by NI elderberry BIB 143.6 mg GAE/100 g). Addition of BIBs to U-CC increased TPC content and DPPH- (2,2-diphenyl-1-picrylhydrazyl)-radical scavenging activity of the U-CC (the highest TPC content was found in C-RaNI 184.5 mg/100 g, and strong positive correlation between TPC and DPPH- of the U-CC was found, r = 0.658). Pre-dominant fatty acids group in U-CC was saturated fatty acids (SFA), however, the lowest content of SFA was unfolded in C-EldAI samples (in comparison with C, on average, by 1.6 times lower). The highest biogenic amine content was attained in C-EldAI (104.1 mg/kg). In total, 43 volatile compounds (VC) were identified in U-CC and, in all cases, broader spectrum of VCs was observed in U-CC enriched with BIBs. After 10 days of storage, the highest enterobacteria number was in C-BluNI (1.88 log10 CFU/g). All U-CC showed similar overall acceptability (on average, 8.34 points), however, the highest intensity of emotion “happy” was expressed by testing C-EldNI. Finally, the BIBs are prospective ingredients for U-CC enrichment in sustainable manner and improved nutritional traits.

Reviewer 3:

Materials and methods section

2.3.1. Evaluation of the antimicrobial properties in berry industry by-products (BIB)

Please cite the standard method followed. Authors didn’t mention the quantity of liquid sample used during the concerned assay. Please add.

Authors response 1: Authors are thankful for comment, information was included:

Wells of 6 mm in diameter were punched in the agar and filled with 50 µL of the tested BIB.

Authors response 2: Authors are thankful for comment, and the reference to the antimicrobial testing method was added. Agar-well diffusion method is not standardized but it is widely used to evaluate the antimicrobial activity of plants, microbial extracts and other natural products (Balouiri et al., Valgas et al). Information was included:

The antimicrobial activity of the BIBs was assessed by measuring the diameter of inhibition zones (DIZ, mm) in agar-well diffusion assays as described previously by Balouiri et al. [24].

Reference:

Balouiri M, Sadiki M, Ibnsouda SK. Methods for in vitro evaluating antimicrobial activity: A review. J Pharm Anal. 2016 Apr;6(2):71-79. doi: 10.1016/j.jpha.2015.11.005. Epub 2015 Dec 2. PMID: 29403965; PMCID: PMC5762448.

Valgas C., De Souza S.M., Smânia E.F.A. Screening methods to determine antibacterial activity of natural products. Braz. J. Microbiol. 2007;38:369–380.

Reviewer 3:

2.3.3. Determination of total phenolic compounds (TPC) and 2, 2-diphenyl-1-picrylhydrazyl (DPPH- )-radical scavenging activity

Information regarding the concentration of sodium carbonate used during the assay is missing in the concerned section.

Further, information related to DPPH reagent has not been mentioned. What about the use of extraction phase/solvent during the assay which was used for dissolving DPPH? What was the concentration of DPPH used? Absorbance details are also missing.

Authors response: Authors are thankful for comment, information was included:

The TPC content of the BIBs was determined by a spectrophotometric method described by Vaher et al. [24]. A total of 0.2 mL of every fraction of free phenolics was blended with 1 mL of Folin–Ciocalteau reagent and 0.8 mL of a saturated sodium carbonate (Na2CO3) solution (7.5%). The prepared mixed solution was stored at room temperature (24 °C) for 30 min in the dark, and, the absorbance was measured at a wavelength of 765 nm with a V-1100D spectrophotometer (J.P. Selecta S.A., Barcelona, Spain). The TPC content was expressed as mg of gallic acid equivalent mL of solution [mg GAE/100 g (DM)] [24]. The ability of the BIB extract to scavenge DPPH- free radicals was assessed using the method described by Zhu et al. [25]. The 400 µl of sample or ethanol (blank) were added to 3600 µl of a 100 µM DPPH ethanolic solution and mixed. Then, after 20 min of storage in the dar at room temperature (24 °C), the absorbance was measured at a wavelength of 517 nm with a V-1100D spectrophotometer (J.P. Selecta S.A., Barcelona, Spain). All measurements were performed in triplicate.

Reviewer 3:

Authors represent the observations from their results but didn’t compare with the latest scientific reports. Further, reason behind the specific observations has not been mentioned. Please clarify.

Authors response: Authors are thankful for comment. Discussion was improved (marked in yellow). However, authors would like to explain that to the best of our knowledge, unripened curd cheese with berry by-products such as those included in our study has not been widely tested by other researchers. Therefore, we believe that all available research data was discussed and obtained results were compared with existing ones in this manuscript.

Reviewer 4 Report

Introduction: Provide a concise overview of the importance and relevance of the topic, including the potential benefits of incorporating raspberry, blueberry, and elderberry by-products into cheese production.

Research Objectives: Clearly define the research objectives or hypotheses of the study. State what you aim to achieve and the specific questions you seek to answer through your research.

Methodology: Sufficient

Experimental Design: Provide details on the replication of experiments and any randomization or blinding procedures implemented.

Analysis and Results: Sufficient material

Discussion and Interpretation: Discuss any significant findings, trends, or relationships observed. Compare your results with existing literature and explain how your study contributes to the field.

Limitations and Future Directions: Suggest avenues for future research that could build upon your findings and address any limitations.

Conclusion: Provide a concise and well-summarized conclusion that emphasizes the key findings of your study. Highlight the implications and potential applications of your research in the context of the cheese industry. Remember to proofread your paper for grammar, spelling, and clarity. Ensure that your writing is concise, logical, and flows smoothly from one section to another

Remember to proofread your paper for grammar, spelling, and clarity. Ensure that your writing is concise, logical, and flows smoothly from one section to another

Author Response

Dear Reviewer, below You will find Authors responses.

Reviewer 4:

Introduction: Provide a concise overview of the importance and relevance of the topic, including the potential benefits of incorporating raspberry, blueberry, and elderberry by-products into cheese production.

Authors response: Authors are thankful for comment, information was highlighted:

Industrial processes, such as juice and wine production, greatly contribute to BIB production. These kinds of residues are generally thrown away in form of leftover and used as feedstock or are composted. However, they are a great source of bioactive compounds (polyphenols, vitamins, minerals, etc.) [8], which are used not efficiently used so far. In this research study, we hypothesized that raspberries (Ras), blueberries (Blu) and elderberries (Eld) production by-products can be employed in high-added value (better antioxidant properties, higher sensory acceptability, due to a higher variety of volatile compounds, etc.) U-CC preparation.

Reviewer 4:

Research Objectives: Clearly define the research objectives or hypotheses of the study. State what you aim to achieve and the specific questions you seek to answer through your research.

Authors response: Authors are thankful for comment, we would like to explain, that the hypotheses of the study was included:

Industrial processes (juice, wine, etc. production) greatly contribute to BIB production, which , till now, are used not enough efficient [8]. In this research study, we hypothesized that raspberries (Ras), blueberries (Blu) and elderberries (Eld) production by-products can be employed in high-added value (better antioxidant properties, higher sensory acceptability, due to a higher variety of volatile compounds, etc.) U-CC preparation.

 as well as the aim:

Finally, the aim of this study was to apply BIB for U-CC enrichment. To implement such a goal, a two stages experiment was performed. During the first stage, antimicrobial properties of the BIBs were tested and the effects of the immobilization technology on BIBs antioxidant properties and color coordinates were evaluated. During the second stage, non-immobilized (NI) and agar-immobilized (AI) BIBs were applied for U-CC enrichment, and their influence on U-CC acidity parameters, color characteristics, moisture content, VC and FA profiles, biogenic amine (BA) concentration, sensory properties, induced emotions for consumers and microbiological characteristics during the storage were analyzed.

Reviewer 4:

Methodology: Sufficient

Authors response: Authors are thankful for evaluation.

Reviewer 3:

Experimental Design: Provide details on the replication of experiments and any randomization or blinding procedures implemented.

Authors response: Authors are thankful for comment, in Section 2.5. is included:

…. (for BIB antimicrobial properties and physicochemical parameters, n = 3; for U-CC physicochemical and microbiological parameters, n = 3; for U-CC overall acceptability and emotions induced for consumers, n = 10).

….. The normal distribution of data was checked using Descriptive Statistics tests. In order to evaluate the influence of the different type of BIBs and immobilization on the analyzed U-CC parameters, data were evaluated by the multivariate analysis of variance (ANOVA) procedure, and Tukey’s honestly significant difference (HSD) procedure as post-hoc tests.

Additionally in section 2.4.6. exclusion criteria are described:

…. Ten judges were recruited internally (Institute of Animal Rearing Technologies and Department of Food Safety and Quality, Lithuanian University of Health Sciences, Kaunas, Lithuania): 5 females and 5 males, from 25 to 50 years old [34,35]. Individuals who were familiar with this study were excluded from the panel. The previous training of the judges was based on descriptive analysis [36–38]. Selected judges were non-smokers, interested in sensory analysis and motivated to participate.

Reviewer 4:

Analysis and Results: Sufficient material

Authors response: Authors are thankful for evaluation.

Reviewer 4:

Discussion and Interpretation: Discuss any significant findings, trends, or relationships observed. Compare your results with existing literature and explain how your study contributes to the field.

Authors response: Discussion was improved (marked in yellow). However, authors would like to explain that to the best of our knowledge, unripened curd cheese with berry by-products such as those included in our study has not been widely tested by other researchers. Therefore, we believe that all available research data was discussed and obtained results were compared with existing ones in this manuscript. Our study provides beneficial data on berry by-product applications for functional unripened cheese manufacture in a sustainable manner.

Reviewer 4:

Limitations and Future Directions: Suggest avenues for future research that could build upon your findings and address any limitations.

Authors response: Authors are thankful for comment, information was included:

         Finally, this study showed that BIBs are prospective ingredients for U-CC enrichment in sustainable manner. However, further research is needed to evaluate possible contamination of the BIBs, to avoid non-desirable changes during the U-CC production, including microbial contamination and BA formation. Also, future research can be applied to the more detailed analysis of the broader spectrum of BIBs, with the aim to use these valuable by-products in dairy industry.

Reviewer 4:

Conclusion: Provide a concise and well-summarized conclusion that emphasizes the key findings of your study. Highlight the implications and potential applications of your research in the context of the cheese industry. Remember to proofread your paper for grammar, spelling, and clarity. Ensure that your writing is concise, logical, and flows smoothly from one section to another

Authors response: Authors are thankful for Reviewer comment, conclusions were corrected:

This study confirmed that the raspberry, blueberry, and elderberry BIBs possess desirable antimicrobial and antioxidant activities and the addition of these by-products to the main U-CC formula increased the TPC content and DPPH- radical scavenging activity of the U-CC. Also, despite that predominant FAs in U-CC were saturated, addition of BIBs led to lower saturated FAs content in the final product. As well as, BIBs increased the number of VCs in U-CC, and these changes are associated with the higher intensity of emotion “happy” induced for consumers by the tested product. However, after 10 days of storage, the highest total enterobacteria number was found in C-BluNI samples. Finally, this study showed that BIBs are prospective ingredients for U-CC enrichment in sustainable manner. However, further research is needed to evaluate possible contamination of the BIBs, to avoid non-desirable changes during the U-CC production, including microbial contamination and BA formation. Also, future research can be concentrated to more detailed analysis of the broader spectrum of BIBs, with the aim to use these valuable by-products in dairy industry.
